# Role of Pcdh15 in the development of intrinsic polarity of inner ear hair cells

**Raman Kaushik**[1], **Shivangi Pandey**[1], **Anubhav Prakash**[1,2], **Fenil Ganatra**[1], **Takaya Abe**[3], **Hiroshi Kiyonari**[3], **Raj K. Ladher**[1]*

**1** National Centre for Biological Sciences, Tata Institute of Fundamental Research, Bangalore, Karnataka, India, **2** Trivedi School of Biosciences, Ashoka University, Sonipat, Haryana, India, **3** Laboratory for Animal Resources and Genetic Engineering, RIKEN Center for Biosystems Dynamics Research, Kobe, Hyogo, Japan

* rajladher@ncbs.res.in

## Abstract

In vertebrates, auditory information is transduced in the cochlea by mechanosensory hair cells (HC) through an eccentrically organised structure known as the hair bundle. This consists of a true cilium, known as the kinocilium, and modified microvilli, known as stereocilia. The hair bundle has a distinct structure with stereocilia organised in graded rows, with the longest abutting the kinocilium. The hair bundles of all HC are aligned to the tissue axis and are planar polarised. Important in the development and physiology of HC are protein bridges consisting of cadherin-23 (CDH23) and protocadherin-15 (PCDH15). These link the tips of stereocilia, where they play a role in mechanotransduction, and between the kinocilia and the stereocilia, where they are involved in development. Both *Cdh23* and *Pcdh15* mutations result in defects in planar polarity; however, the mechanism through which this defect arises is unclear. Using a novel mutant for the *Pcdh15-CD2* isoform, we show that while the initial deflection of the kinocilium occurs, its peripheral migration to register with Gαi is perturbed. *Pcdh15-CD2* genetically interacts with *Gpsm2*, perturbing vestibular function. We find that the earliest expression of PCDH15-CD2 is at the base of the kinocilia, and the defects in morphogenesis occur before the formation of kinocilial links. By re-introducing functional PCDH15-CD2, we show that polarity can be restored. Our data suggest that, in addition to its adhesive role, PCDH15-CD2 has an early role in intrinsic hair cell polarity through a mechanism independent of kinocilial links.

## Author summary

Hearing, like other sensory modalities, is essential for animals to perceive and interact with their environment. The cochlea, the auditory sensory organ, develops under tight genetic regulation. In this study, we investigate the role of *Pcdh15*, a gene previously implicated in maintaining hair bundle integrity in

**Data availability statement:** All raw data is provide as Supporting information with the manuscript.

**Funding:** RKL received funding from Department of Atomic Energy, Government of India, Project Identification No. RTI 4006. RK, SP, FG and RKL received a salary or stipend from this grant. RKL received funding from Royal National Institute for Deaf People G97, SERB CRG and Infosys Foundation through a TIFR Infosys-Leading Edge Grant AP: received a Simons Foundation: Simons Ashoka Early Career Fellowship. AP received a salary from this grant. The funders had no role in study design, data collection and analysis, decision to publish, or preparation of the manuscript.

**Competing interests:** The authors have declared that no competing interests exist.

hair cells. Using mouse genetics and advanced microscopy, we uncover an extended role for *Pcdh15-CD2* isoform in regulating intrinsic hair cell polarity. Our findings reveal that *Pcdh15-CD2* functions not merely in forming physical links between kinocilium and stereocilia but plays a key step in establishing intrinsic hair cell polarity. Prior to forming structural connections, *Pcdh15-CD2* contributes to the directionality of kinocilium positioning. Importantly, we demonstrate that restoring *Pcdh15-CD2* expression in mutant mice rescues hair cell polarity in a time-sensitive manner, highlighting its dynamic and instructive role in cochlear development.

## Introduction

Hearing has conferred numerous evolutionary advantages to animals, such as prey detection, predator avoidance and communication. Sound is sensed in the cochlea of the inner ear, in a specialised epithelium known as the organ of Corti (OC). OC comprises four rows of mechanosensory hair cells (HC), divided into three rows of outer (OHC) and one row of inner hair cells (IHC), all surrounded by supporting cells [1,2]. HC have hair bundles on their apical surface, which consist of graded rows of actin-based stereocilia increasing in height, with the tallest attached to a single true cilium, the kinocilium [3]. The development and orientation of the hair bundles are critical for the functioning of the inner ear as they are only mechanosensitive along one axis, the short-tall axis defined by the kinocilium [4]. Hair bundle development starts with a centrally positioned kinocilium surrounded by microvilli covering the apical surface [5–8]. The kinocilium then centrifugally repositions to the periphery of the hair cell, followed by a re-orientation along the edge towards the abneural side of the hair cells [9,10]. Microvilli near the kinocilium then elongate and form the stereocilia, and the whole hair bundle moves towards the centre of the apical surface [11]. These initial steps of hair cell development define its intrinsic polarity. This intrinsic polarity is aligned across the OC such that the intrinsic polarity of all HC is aligned to the tissue axis, a process called planar cell polarity (PCP) [12].

Mutation studies have revealed a complex interplay of genes and pathways governing these processes. PCP is controlled by a molecularly conserved pathway of genes. In the cochlea, these comprise at least six different core PCP proteins [11,12]. These proteins are expressed asymmetrically at the intercellular junctions and are crucial for the tissue-wide arrangement of hair cells. Intrinsic polarity of HC relies on the interaction of the kinocilium and a signalling complex initially defined in spindle positioning during asymmetric cell division. These proteins include the inhibitory G protein subunit alpha (Gαi), its binding partner GPSM2 (G protein signalling modulator) and Inscuteable (INSC) [13]. This interaction establishes the apical architecture of the hair cell. Mutants of any of these genes result in hair cells with varying defects in intrinsic polarity, ranging from a failure of the lateral-ward movement, hair bundle fragmentation, and decoupling of the kinocilium from the stereociliary bundle [9,10,14,15]. These mutants also show mild PCP defects, with increased deviation

from the tissue axis. Similar defects have also been observed in mutants of genes involved in various aspects of cilia function, such as those of intraflagellar transport complex components or of molecules involved in kinocilia assembly [16–20]. Perturbations in the localisation of Gαi protein in the kinocilia mutants, *Mkss*, *Bbs8* or *Ift88,* suggest a relationship between the kinocilia and the signalling mechanisms that are thought to underlie apical patterning of HC [10,14,18].

Protein links between its components ensure the cohesion of the hair bundle [21]. These include links between the stereocilia and between the kinocilium and stereocilia. Studies on the components of the kinocilial links, CDH23 and PCDH15, suggest a function of these molecules in the generation of HC polarity [22,23]. *Pcdh15* has three major isoforms (PCDH15-CD1, PCDH15-CD2, and PCDH15-CD3) that differ from each other in their cytoplasmic domains. They are expressed through the mutually exclusive alternative splicing of exons 35, 38, and 39, respectively [24–27] (Fig 1A). Among these isoforms, only mutants of the CD2 isoform show mild planar polarity defects. In addition, these mutants also display a perturbation in the apical architecture of the hair cell, showing a decoupling of the kinocilium from the tallest stereocilia [28]. Blocking FGF signalling prevents PCDH15 from entering the kinocilium, which also results in kinocilium-stereocilia decoupling [29]. However, kinocilial links only appear after the off-centre relocation of the kinocilia [30]. This may suggest a role for PCDH15, independent of its function in kinocilial links. In this study, we generate a mouse where *Pcdh15-CD2* is fused to an epitope tag. We find that PCDH15-CD2 protein is initially localised around the base of the cilium prior to the peripheral relocation. Mutation of *Pcdh15-CD2* alters the localisation of Gαi, similar to kinocilial mutants. We find evidence of a genetic interaction with *Gpsm2*, and in mice mutant for both *Gpsm2* and *Pcdh15-CD2*. Here not only exaggerate polarity defects and bundle fragmentation, we now find defects in vestibular function. By re-expressing *Pcdh15-CD2,* we find that it is possible to rescue these defects up to 2 days after peripheral relocation. Our data suggest an additional role for PCDH15-CD2 during the generation of intrinsic hair cell polarity in coupling the kinocilium to the domain of GPSM2-Gαi signalling.

## Methods

### Ethics statement

Experiments on animals were performed after approval from the NCBS Institutional Animal Ethics Committee (Approval no. NCBS-IAE-2023/10 (R2)).

### Animal use and care

*Pcdh15-n38* mice (Accession No.: CDB0186E: https://large.riken.jp/distribution/mutant-list.html) were generated by CRISPR/Cas9-mediated knock-in in zygotes as previously described [31]. Briefly, the following guide RNAs (gRNAs) were designed by using CRISPRdirect [32] to insert the knock-in construct: 5' gRNA target sequence – 5'-CTATATGCATAGCTTGCGTG-3' and 3' gRNA target sequence - 5'-CTCAGCAAGCACGTCTACGT. Founder mice were bred with C57BL/6N mice to generate heterozygous animals. *Pcdh15-n38YF* mice were made by crossing homozygous *Pcdh15-n38* with *Pgk1-Cre* ([33]: JAX#020811) mice. *Gpsm2* mice (Accession No. CDB1144K) were received from RIKEN BDR, and were the kind gift of Fumio Matsuzaki [34]. *Atoh1-Cre/ESR1* (JAX#007684) mice were obtained from JAX [35]. All primers used for genotyping are detailed in the table (S1 Table).

### Immunofluorescence

Inner ear dissections were performed as previously described [36,37]. Fixation was based on the choice of antibody used (S2 Table). Fixed inner ears were washed with 1X PBS and further dissected to expose the sensory epithelium. Tectorial membrane was removed before permeabilisation with PBS + 0.3%Tween-20. Tissues were blocked with blocking buffer (10% goat serum, 1% bovine serum albumin (BSA) in permeabilisation buffer) and then incubated with primary antibodies (S2 Table) overnight at 4°C. Tissues were then washed 5–6 times with PBST before incubating with secondary antibodies

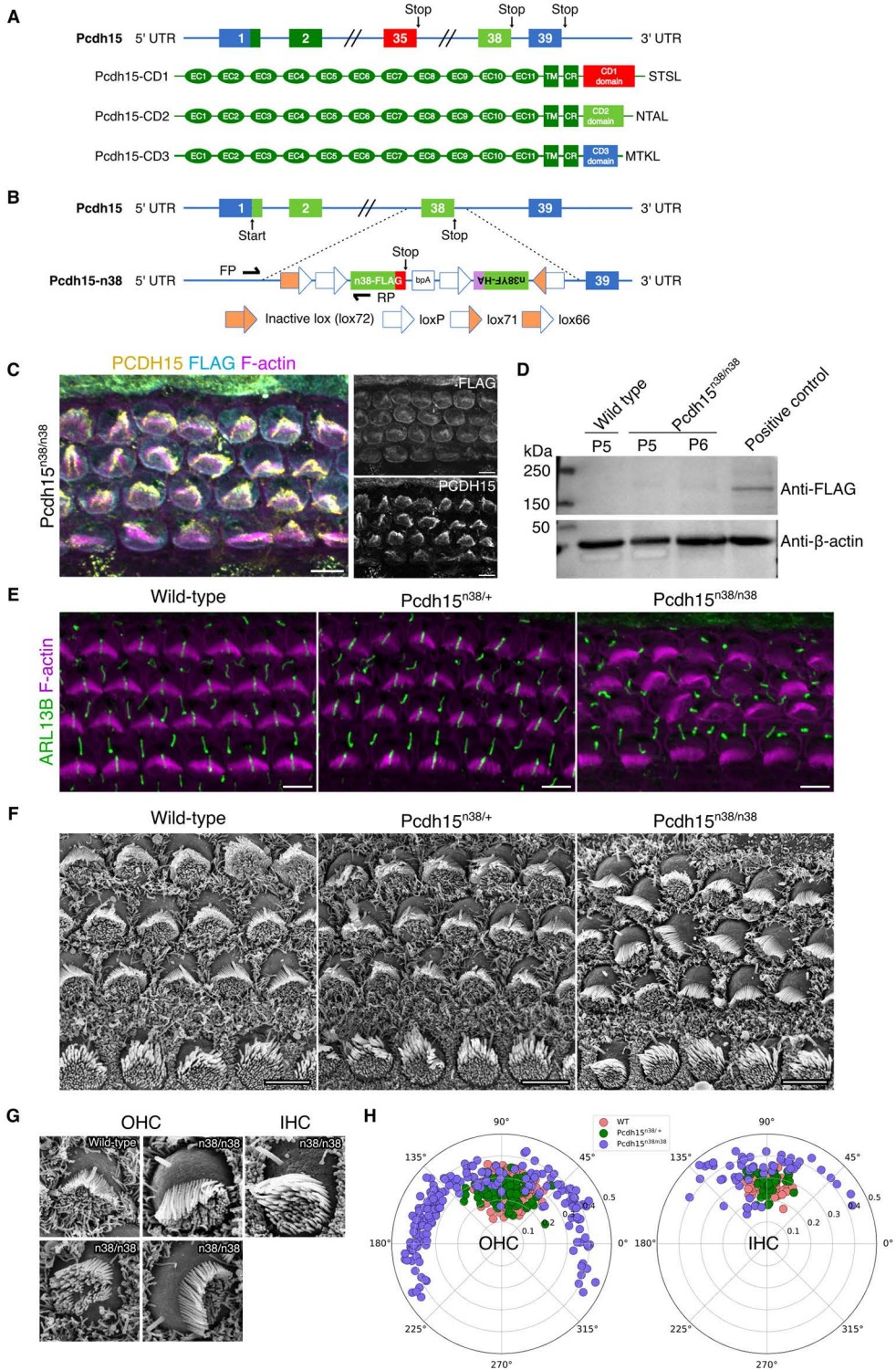

**Fig 1. *Pcdh15^{n38/n38}* mice are functionally null and act similarly to *Pcdh15-CD2* mutants.** (A) Schematic of the *Pcdh15* exons that produce three iso-forms, CD1, CD2 and CD3, by alternative splicing of exons 35, 38 and 39, respectively. (B) Schematic of the DNA construct used to generate *Pcdh15^{n38/n38}* mice. (C) The organ of Corti of *Pcdh15^{n38/n38}* mice at P0 (mid). Immunostaining fails to detect the PCDH15-CD2-FLAG signal (cyan), although endogenous PCDH15 (yellow) is localised at the tip of stereocilia (magenta, marked with phalloidin). (Scale bar = 5 μm). (D) Western blot fails to detect

PCDH15-CD2-FLAG protein in *Pcdh15$^{n38/n38}$* mice OC (P5/6). Positive control shows HEK293T cells lysate transfected with PCDH15-CD2-FLAG cDNA. β-actin is used as a loading control. (E-F) Fluorescence (E) and SEM (F) imaging show hair bundle polarity defects in *Pcdh15$^{n38/n38}$* OC when compared to normal hair bundle orientation in WT & *Pcdh15$^{n38/+}$* (P0, mid). Cilia are marked with ARL13B (green) and stereocilia with phalloidin (magenta). (Scale bar = 5 μm). (G) Kinocilium is dissociated from stereocilia in OHC and IHC of *Pcdh15$^{n38/n38}$* OC when compared to WT. (H) Polar plots of kinocilia positions in hair cells show higher circular standard deviation, indicative of hair bundle polarity defects in *Pcdh15$^{n38/n38}$* OC (slate blue). This is much reduced in WT (pink) & *Pcdh15$^{n38/+}$* (green) (P0, mid). (Sample size for OHC/IHC, WT = 156/51, *Pcdh15$^{n38/+}$* = 119/40, *Pcdh15$^{n38/n38}$* = 184/57).

and phalloidin (S3 Table) at room temperature for 1 hour. Tissues were washed 5–6 times before mounting with aqueous mounting media.

## Image acquisition and processing

Confocal images were captured by an Olympus FV3000 inverted confocal microscope with a 60x objective (1.42 NA) using its software Olympus FV31S-SW. All images were captured at sub-saturation level under Nyquist sampling criteria. High-resolution images were taken on a Zeiss LSM 980 Airyscan 2 microscope with a 63x objective (1.4 NA). All the raw files were processed by ImageJ software.

## Immunoprecipitation and western blotting

For immunoprecipitation (IP), 20 mouse cochleae at P5 stage were dissected and homogenised in lysis buffer (0.5% Triton X-100, 150mM NaCl, 5mM EDTA and 1X MS-SAFE protease and phosphatase inhibitor in PBS). Lysates were cleared by centrifugation at 15,000 g for 10 mins. Anti-HA or normal mouse IgG antibody was incubated with Dynabead Protein G beads (Invitrogen) for 1 hr at 4°C to form the Dynabead-bound antibody complexes. Parallelly, the lysate was incubated with Dynabeads to reduce the non-specific binding. The cleared lysates were incubated with Dynabead-bound antibody complexes of either HA-specific antibody (Sigma H3663) or with normal mouse IgG overnight at 4°C. The proteins were purified and washed 5 times with lysis buffer before eluting in 1X SDS loading buffer. Proteins were resolved by 10% SDS-PAGE and transferred onto PVDF membranes. The membranes were blocked with 5% BSA in 0.1% TBST buffer and incubated with primary antibodies overnight at 4°C. After 5 washes with 0.1% TBST buffer, membranes were incubated with peroxidase-conjugated secondary antibodies. Immunoreactive proteins were detected with Western Bright ECL detection reagents.

## Vestibular test

Littermate mice (5 months old) of different genotypes (*Pcdh15$^{n38/+}$:: Gpsm2$^{+/-}$*, *Pcdh15$^{n38/n38}$:: Gpsm2$^{+/-}$*, *Pcdh15$^{n38/+}$:: Gpsm2$^{-/-}$* and *Pcdh15$^{n38/n38}$:: Gpsm2$^{-/-}$*) were kept in a controlled environment. Vestibular test experiments were performed in a box (22cm X 15 cm X 13 cm), and the activity of mice was recorded for 5 min. The total distance covered, mean speed and no. of rotations were measured. Analyses were performed using ANY-maze software.

## Preyer's reflex test

Adult mice (3 months old) of different genotypes (wild-type, Pcdh15$^{n38/n38}$ and Pcdh15$^{n38YF/n38YF}$) were kept in a controlled environment. Preyer's reflex test was performed in a box (22cm X 15 cm X 13 cm), and the startle response was recorded for 10 stimuli (spaced by 2 min). The presence of the startle response was measured as a percentage of positive Preyer's reflex.

## Image analysis and statistics

Data were collected from at least three animals (from at least two different breeder pairs) for each experiment. The sample size is mentioned with each individual graph. Images were analysed using ImageJ and Imaris.

For hair bundle defects, kinocilium position (marked with ARL13B or acetylated tubulin antibodies) was measured with respect to the hair cell cortex (marked by phalloidin). A vector was drawn from the centre of the hair cell to the kinocilium base, whose length (r) is normalised with cell radius and the angle (θ) from the proximal to distal (P-D) axis. The kinocilium position in different hair cells was visualised by using polar plots in the Matplotlib library of Python. Similarly, the frequencies of kinocilium orientation were measured and plotted from 0°-360° within 10° brackets. For measurement of stereocilia bundle orientation, the angle tool in ImageJ software was utilised, where a line was drawn from the centre of the stereocilia bundle and then plotted with the tissue axis.

For spatial expression measurement of HA staining, a line (ROI) of width equal to the kinocilium (marked by acetylated tubulin) was taken, and the fluorescence intensity of acetylated tubulin and HA was measured with respect to the distance (on the line drawn). The fluorescence intensity of both signals was plotted together to give their colocalisation pattern.

For alpha – theta (α-θ) correlation analysis, θ angles were plotted against α angles. θ angles were calculated for the angle of the kinocilia base from 0° (along the P-D axis), and α angles were calculated for the centre of the Gαi expression domains from the horizontal (0°) on the P-D axis. The Python script is available as supporting information.

Box and whisker graphs were plotted using GraphPad Prism 9.5.0. The same software was used for statistical analyses. We analysed data using ordinary one-way or two-way ANOVA test (Tukey's multiple comparison test). P values were represented in the graphs; $p < 0.05$ was considered statistically significant.

### Scanning electron microscopy (SEM)

Samples for SEM were prepared as described previously [37]. After inner ear dissection in 1X PBS, tissues were fixed in 2.5% glutaraldehyde in sodium cacodylate buffer (0.1M) with calcium chloride (3mM) for 48 hours. After fixation, samples were micro-dissected to expose the sensory epithelium. Samples were again processed with the OTOTO method for an alternate series of fixation with 1% osmium tetroxide (O) and 0.5% thiocarbohydrazide (T). Samples were dehydrated first with serially increased alcohol percentage, followed by critical point dehydration (Leica EM CPD300). Mounting was done on double-sided carbon adhesive tapes, followed by sputter coating. Samples were imaged on Zeiss Merlin Compact VP with SE2 detectors.

## Results

### Generation of $Pcdh15^{n38/n38}$ and $Pcdh15^{n38YF/n38YF}$ mice

Our previous work had suggested that tyrosine phosphorylation on the cytoplasmic domain of the PCDH15-CD2 isoform was important for PCDH15 function [29]. We hypothesised that finding interactions specific to phosphorylation of PCDH15 would help understand this function. To do this, we engineered a mouse where the endogenous exon 38 was replaced by a construct that consisted of a CD2 domain with an in-frame FLAG tag. In addition, the construct contained a mutant CD2 in which all 6 tyrosine residues were changed to phenylalanine and fused to an HA tag. These sequences were flanked by *LoxP*, *Lox71* and *Lox66* sites. Without Cre recombination, the wild-type PCDH15-CD2 tagged with the FLAG epitope should be expressed ($Pcdh15^{n38/n38}$) (Figs 1B and S1A).

To check for PCDH15-CD2 expression in hair cells of the organ of Corti, we performed immuno-localisation with a FLAG-tag specific antibody in $Pcdh15^{n38/n38}$ mice. While PCDH15 expression could be visualised in stereocilia tips, we were unable to detect immuno-reactivity for the FLAG-tag (Fig 1C). The lack of FLAG immunoreactivity in the tissue was confirmed by Western blotting. We failed to detect the FLAG epitope in P5/6 $Pcdh15^{n38/n38}$ organ of Corti lysates, despite detection in HEK cells (Figs 1D and S1B). RT-PCR indicated no difference in mRNA expression between WT and $Pcdh15^{n38/n38}$ mice (S1C Fig). This suggested that translation of the mutant PCDH15-CD2 isoform was impaired, indicating that the $Pcdh15^{n38/n38}$ mice are functionally null and act similar to $Pcdh15$-ΔCD2 mutants reported earlier [28,38].

## Defects in hair bundle polarity in Pcdh15$^{n38/38}$ mice

Pcdh15-ΔCD2 mutants show defects in hair bundle orientation and mechanotransduction [28,38]. To ask if Pcdh15$^{n38/n38}$ mice show a similar phenotype to Pcdh15-ΔCD2, we immunostained the P0 organ of Corti with ARL13B and phalloidin to determine hair bundle polarity. The average kinocilium deviation from the tissue axis was comparable in P0 wild-type and Pcdh15$^{n38/+}$ animals (Fig 1E, 1F and 1H). In OHC, this deviation, at P0, is 13˚±11˚ in Pcdh15$^{n38/+}$; however, in Pcdh15$^{n38/n38}$ animals, this deviation is 59˚±32˚, indicating a defect in planar polarity. While hair bundle polarity is affected equally in all three rows of OHC, IHC show a less severe defect in hair bundle polarity, with the average kinocilium deviation of 23˚±15˚ in Pcdh15$^{n38/n38}$ as compared to 8˚±6˚ in Pcdh15$^{n38/+}$ animals (Fig 1H). Despite the defect in polarity, the expression of the core PCP protein, VANGL2, is unchanged in Pcdh15$^{n38/n38}$ mice (S1D Fig). Closer inspection using SEM revealed a decoupling of the kinocilium from the stereocilia (Fig 1F). In Pcdh15$^{n38/+}$, almost all kinocilia are attached to the longest stereocilia and are found at the vertex of the stereocilia bundle. In Pcdh15$^{n38/n38}$ mice, we were unable to detect any hair cell with a kinocilium coupled to the stereocilia. Additionally, we observed misoriented stereocilia bundles of varying degrees from the tissue axis, as well as occasional circular bundles with a frequency <1% (Fig 1G). Given that the kinocilium is decoupled from the stereocilia bundle in the mutants, we quantified orientation defects by measuring the angular deviation of stereocilia relative to the tissue axis. The average stereocilia bundle deviation, 26˚±12˚ in Pcdh15$^{n38/38}$ mice OHC is significantly higher when compared to 15˚±8˚ in Pcdh15$^{n38/+}$ controls (S1E Fig).

## Tyrosine phosphorylation of the CD2 domain is not essential for PCDH15 function in mice

Our previous work had identified that FGFR1 signalling was involved in the coupling of the kinocilium to the stereocilia, through the phosphorylation of the cytoplasmic domain of PCDH15-CD2 [29]. To generate the Pcdh15$^{n38YF/+}$ germline mutant, we mated the Pcdh15$^{n38/n38}$ mice with the ubiquitously expressed Pgk1-Cre. The Pcdh15$^{n38YF/+}$ heterozygous animals were back-crossed to give homozygotes (Figs 2A, 2B and S2A). These mice showed specific epitope tag (HA) expression in the kinocilium and at stereocilia tips, indicating normal expression of the modified PCDH15-CD2-HA (Fig 2C, 2D). This expression colocalised with PCDH15 expression, validating the normal expression of the HA-tagged modified PCDH15-CD2 (S2B Fig). Although we could not resolve the hair bundle at embryonic stages, at postnatal stages, we observed PCDH15-CD2-HA expression at the tip of all three rows of stereocilia in HC (S2C Fig). This data was verified by immunoprecipitation from the cochlear lysate of Pcdh15$^{n38YF/n38YF}$ mice, showing a specific band when compared to the IgG control (Figs 2E and S2D). ARL13B and phalloidin staining were used to measure hair bundle polarity, together with SEM for determining kinocilium-stereocilia coupling. Despite Pcdh15$^{n38/n38}$ showing defects in polarity, Pcdh15$^{n38YF/n38YF}$ were similar to wild-type controls (Figs 2F–2H and S2E). This suggests that the phosphorylation of tyrosine in the CD2 domain of Pcdh15$^{n38YF/n38YF}$ is not essential for normal PCDH15 function.

PCDH15-CD2 has also been implicated in mechanotransduction, although the MET channels in Pcdh15-ΔCD2 are functional at early post-natal stages [28]. Using the styryl pyridinium dye, FM1–43 FX, which stains hair cells after entering through MET channels, we asked if MET channels were functional [39]. We find that HC in the P5 organ of Corti from both Pcdh15$^{n38/n38}$ and Pcdh15$^{n38YF/n38YF}$ mice are stained by FM1–43, suggesting that MET channels can open (S2F Fig). Using Preyer's reflex, we found that P60 Pcdh15$^{n38/n38}$ mice could not respond to sound, but Pcdh15$^{n38YF/n38YF}$ mice responded similarly to the control animals (S2G Fig). This data suggests that, similar to Pcdh15-ΔCD2 mutants [28,38], Pcdh15$^{n38/n38}$ mice cannot hear.

## PCDH15-CD2 is expressed in the kinocilium prior to its off-centre movement

To understand the role of PCDH15-CD2 in intrinsic HC polarity, we utilised the HA tag on Pcdh15$^{n38YF/n38YF}$ to map its earliest expression during hair bundle morphogenesis. At E15.5, we observe PCDH15-CD2-HA expression at the base of the organ of Corti, in the IHC kinocilium which at these stages is yet to move off-centre (Fig 3A). We also noted HA positive

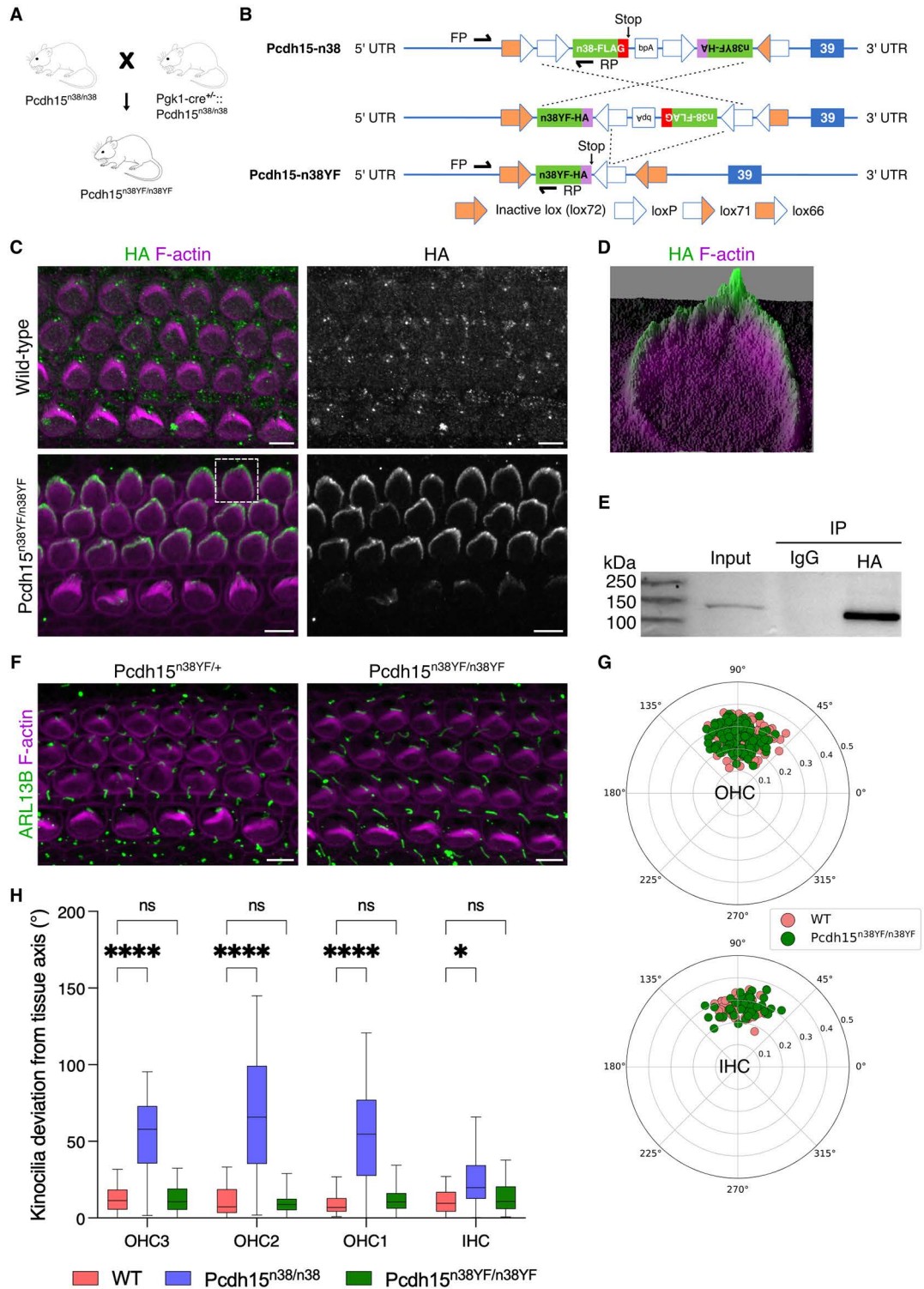

**Fig 2. Tyrosine phosphorylation of the CD2 domain is not essential for PCDH15 function.** (A) Schematic of the mice breeding scheme to produce *Pcdh15^{n38YF/n38YF}* mice. (B) Schematic of the *Pcdh15-n38* DNA construct recombination to *Pcdh15-n38YF* under Cre activity. (C) Modified PCDH15-CD2-HA (green) is expressed in the stereocilia (magenta) of *Pcdh15^{n38YF/n38YF}* hair cells (P0, mid), but not in WT. (Scale bar = 5 μm). (D) Surface plot of OHC shows PCDH15-CD2-HA (green) localisation at the tip of the hair bundle. (E) IP assay shows that PCDH15-CD2-HA is precipitated with anti-HA antibody. (F) Hair bundle polarity is normal in *Pcdh15^{n38YF/n38YF}* and *Pcdh15^{n38YF/+}* hair cells (P0, mid). Cilia are marked with ARL13B (green) and stereocilia

with phalloidin (magenta). (Scale bar = 5 μm). (G) Polar plots marking kinocilia positions in hair cells show normal hair bundle polarity in $Pcdh15^{n38YF/n38YF}$ OC (green) (P0, mid). (Sample size for OHC/IHC, WT = 156/51, $Pcdh15^{n38YF/n38YF}$ = 167/48). (H) Kinocilia are aligned normally with the tissue axis in WT (pink) and $Pcdh15^{n38YF/n38YF}$ (green) hair cells, but significantly deviated in $Pcdh15^{n38/n38}$ (slate blue). The deviation is greater in OHC when compared to IHC (P0, mid). (Scale bar = 5 μm, Two-way Anova with Tukey's multiple comparisons test, ns = P > 0.05, * = P < 0.05, ** = P < 0.01, *** = P < 0.001, **** = P < 0.0001). (Sample size for OHC3/OHC2/OHC1/IHC, WT = 52/52/52/51, $Pcdh15^{n38/n38}$ = 62/60/62/57, $Pcdh15^{n38YF/n38YF}$ = 56/57/54/48).

cortical punctae around the kinocilium (Fig 3A). At these stages, we did not observe HA expression in the OHC or any HC at the apex or mid of the organ of Corti. At E16.5, HA-staining could be observed in the kinocilia of both IHC and OHC at the apex of the organ of Corti (Fig 3B). HA staining was not detected in wild type E16.5 organ of Corti (S2H Fig). To understand the spatial HA expression at the kinocilium, we measured the co-staining of HA with acetylated tubulin, a marker that labels the full length of the kinocilium, including its transition zone [40]. PCDH15-CD2-HA is expressed as punctae along the length of the centrally situated kinocilium of the apex OHC (Fig 3C). In the kinocilium of more mature IHC of the base region, which had migrated eccentrically, expression is also seen in the tip and shaft of the kinocilium but is absent from its base (Fig 3D, 3E).

To ask if PCDH15-CD2-HA was also expressed on the hair bundle of vestibular HC, we assessed the utricle and cristae. At E18.5, newly differentiating hair cells can be distinguished from mature HC. In newly differentiating HC, PCDH15-CD2-HA expression is found in the kinocilium, prior to its off-centre movement. Similar to the auditory HC, PCDH15-CD2-HA expression was also observed in punctae around the HC kinocilium of the utricle (Fig 3F) and cristae at E18.5 (Fig 3G). In more mature HC, distinct punctae in kinocilia are visible. These were not detected in wild type controls (S2I Fig).

### $Pcdh15^{n38/n38}$ mice exhibit early hair bundle polarity defects

During the development of the hair cell, the kinocilium initially moves to an eccentric location, adjusts along the periphery, and then relocalises towards the centroid of the HC apex with the expansion of the bare zone, a region free of microvilli, lateral to the kinocilium [11]. As PCDH15-CD2 expression could be detected in kinocilia before the off-centre movement, we asked at which point polarity defects arose. We determined kinocilia position using ARL13B at E16.5 and E18.5. In $Pcdh15^{n38/+}$ at E16.5, and prior to its off-centre movement, kinocilium position is distributed around the centroid of the HC with a slight bias to the abneural side of the OC. In contrast, the kinocilium does not show such a bias in $Pcdh15^{n38/n38}$ mice (Fig 4A–4C). We next assessed the kinocilia position pattern at E18.5. $Pcdh15^{n38/+}$ kinocilia had relocalised towards the centroid of the HC. In contrast, the kinocilia of $Pcdh15^{n38/n38}$ mice are distributed randomly and remain positioned near the periphery of hair cells (Fig 4D–4F). To further understand the randomised kinocilia position, we investigated the position of centrioles. In $Pcdh15^{n38/+}$ HC at E18.5, centrioles of the basal body are aligned orthogonally, with the mother centrioles attached to the kinocilia, and the daughter centriole extending basally. In contrast, the centrioles are separated from each other in $Pcdh15^{n38/n38}$ mice, and although the mother centriole is still in contact with the kinocilium, the daughter centriole position is randomised (Figs 4G–4I and S3A). Taken together, these data suggest that in $Pcdh15^{n38/n38}$ mice, although the initial kinocilium off-centre movement occurs, its subsequent relocalisation is impaired.

### The coordination of kinocilium with Gαi expression is lost in $Pcdh15-CD2$ mutants

Intrinsic polarity is controlled through a pathway involving GPSM2 and Gαi. Gαi expression is normally restricted to a narrow lateral crescent in HC, marking the bare zone [9,10,14]. In mutants for the ciliary genes $Bbs8$, $Ift88$ or McKusick-Kaufman syndrome ($Mkks$) gene, the expression domain of Gαi is expanded, such that it covers the entire lateral half of the HC [10,14,18]. Similar to ciliary mutants, we find that the loss of $Pcdh15-CD2$ also expands Gαi and GPSM2 expression medially (Figs 5A–5C and S4A–S4C).

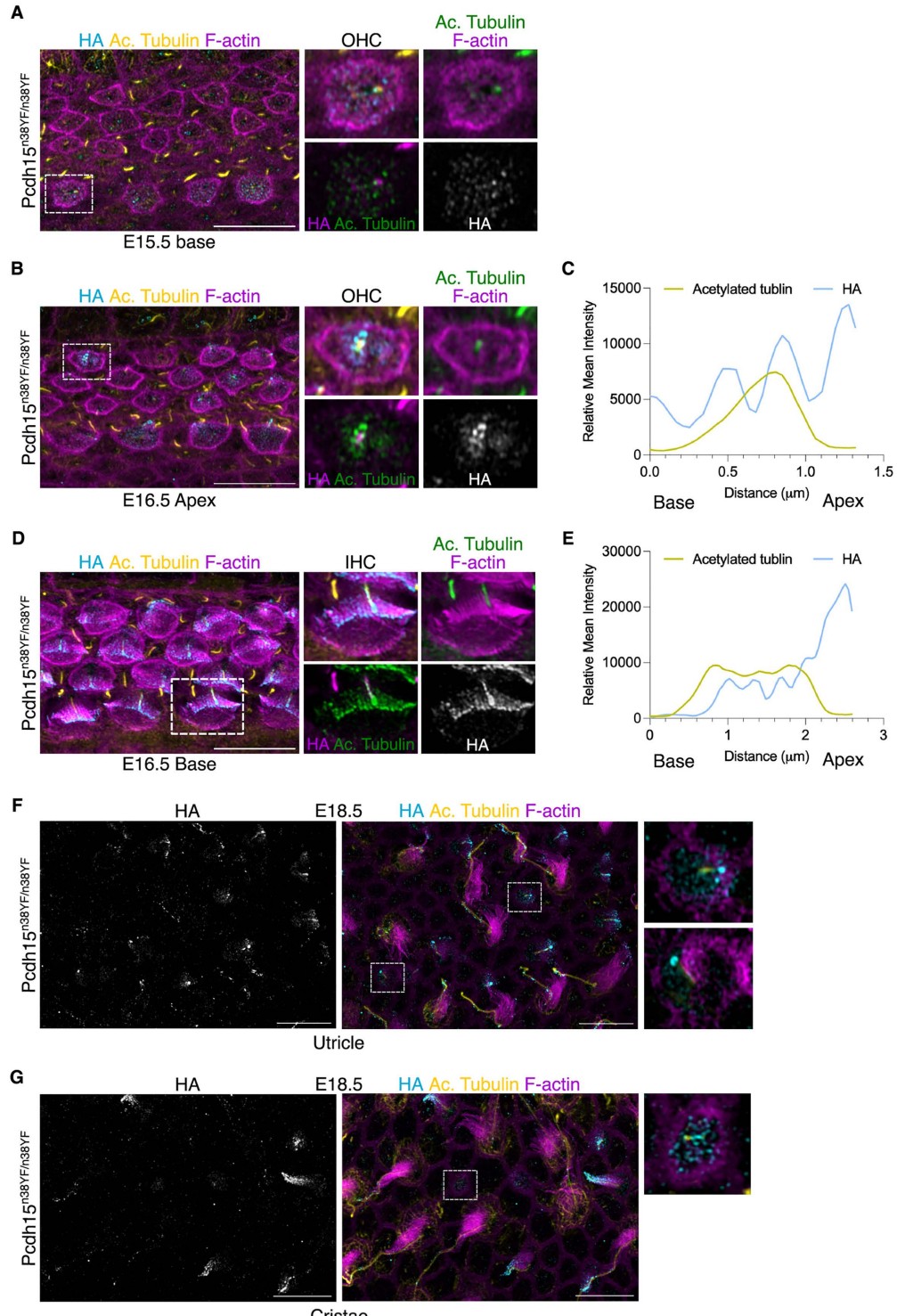

**Fig 3. PCDH15-CD2-HA is expressed in the kinocilium before its off-centre movement.** (A) PCDH15-CD2-HA (cyan) is localised in the centrally located kinocilium (yellow, marked with acetylated tubulin) of IHC of E15.5 OC (base). (Scale bar = 10 µm). (B-C) PCDH15-CD2-HA (cyan) is localised along the entire length of the centrally located kinocilium (yellow, marked with acetylated tubulin) of OHC of E16.5 OC (apex). (Scale bar = 10 µm). (D–E) PCDH15-CD2-HA (cyan) is localised in the laterally positioned kinocilium (yellow, marked with acetylated tubulin) of the IHC of E16.5 OC (base), except

at their basal end. (Scale bar = 10 µm). (F-G) PCDH15-CD2-HA (cyan) is localised in the centrally located kinocilium (yellow, marked with acetylated tubulin) of newly differentiating HC of E18.5 utricle (F) and cristae (G). As hair cells mature, this expression is enriched in hair bundles (magenta, marked with phalloidin) and gets refined. (Scale bar = 10 µm).

After its off-centre movement, the kinocilium undergoes peripheral relocation, such that it is situated in the middle of a domain of Gαi expression. In *Mkks* mutants, this shift is perturbed [10]. To ask if a similar effect was visible in the *Pcdh15-CD2* mutants, we measured the correlation between angle the kinocilium makes with the tissue axis (theta) and the angle that the midpoint of Gαi expression makes with the tissue axis (alpha). In E18.5 *Pcdh15$^{n38/+}$* organ of Corti, a strong angular correlation between centre of the Gαi expression domain and kinocilia is observed, showing a Pearson correlation coefficient of 0.667. In *Pcdh15$^{n38/n38}$* mutant mice, this correlation coefficient is reduced to 0.2 (Fig 5D), suggesting that *Pcdh15$^{n38/n38}$* mutant HC fail to undergo a peripheral relocation during hair bundle development, remaining decoupled from the mid-point of the Gαi domain.

### PCDH15-CD2 and GPSM2 proteins operate cooperatively in hair bundle development

We next asked if PCDH15-CD2 and Gαi-GPSM2 showed any genetic interaction. We used a null *Gpsm2* mutant, in which almost the entire coding sequence is deleted [34]. On a *Pcdh15$^{n38/+}$* background, neither *Gpsm2$^{+/-}$* heterozygotes (10˚ ± 8˚) nor *Gpsm2$^{-/-}$* homozygotes (7˚ ± 6˚) show significant defects in planar polarity at P1 (Fig 6A, 6C and 6E). *Pcdh15$^{n38/n38}$* mice showed mild defects in hair bundle polarity (Figs 1H and 2H), which was similar to both *Pcdh15$^{n38/n38}$:: Gpsm2$^{+/-}$* and *Pcdh15$^{n38/n38}$:: Gpsm2$^{-/-}$* compound mutants (67˚ ± 43˚ and 59˚ ± 39˚, respectively) (Figs 6B, 6D, 6E and S5A).

*Gpsm2* mutant HC show hair bundle fragmentation (24% of OHC & 14% of IHC) (Fig 6F) [9,10,14]. We thus asked if the number of hair cells showing bundle fragmentation increased in the compound mutants. We found increased fragmentation in the compound mutants, with OHC more affected than IHC (46% of OHC & 28% of IHC) (Figs 6F and S5B). Moreover, and in contrast to planar polarity, the proportion of HC in the *Pcdh15$^{n38/n38}$:: Gpsm2$^{-/-}$* organ of Corti that show bundle fragmentation is significantly higher than in *Pcdh15$^{n38/n38}$:: Gpsm2$^{+/-}$* and *Pcdh15$^{n38/+}$:: Gpsm2$^{-/-}$* cochlea (Figs 6F and S5B).

To further characterise the compound mutant phenotype, we asked if we could detect vestibular perturbations. Single *Gpsm2* null and *Pcdh15$^{n38/n38}$* mice do not exhibit vestibular defects, however, we observed that *Pcdh15$^{n38/n38}$:: Gpsm2$^{-/-}$* mice show circling behaviour and hyperactivity (Fig 6G and S1 Movie). In an open field test, we found that *Pcdh15$^{n38/n38}$:: Gpsm2$^{-/-}$* mice travel larger distances, with higher mean speed and show a larger number of rotations, with a preference to anti-clockwise turns (Fig 6H–6J).

To further investigate the vestibular defects underlying the circling behaviour in double mutant mice, we first examined the morphology of hair bundles in the utricle, saccule, and crista using scanning electron microscopy (SEM). Compared to controls, stereocilia bundles in *Gpsm2$^{-/-}$* and *Pcdh15$^{n38/n38}$:: Gpsm2$^{-/-}$* appeared shorter across all three vestibular organs. In contrast, hair bundle morphology in *Pcdh15-CD2* mutant mice was comparable to that of control animals (Fig 7A). In the utricle and saccule of *Pcdh15$^{n38/+}$:: Gpsm2$^{-/-}$* and *Pcdh15$^{n38/n38}$:: Gpsm2$^{-/-}$,* bundle gradation was much reduced. In the cristae, the stereociliary bundle was still graded in *Pcdh15$^{n38/+}$:: Gpsm2$^{-/-}$* mutants. Interestingly, gradation was absent in the cristae of double mutants (Figs 7A and S6A). Although it remains unclear whether this loss of gradation in the cristae alone can account for the circling phenotype, these findings suggest that a cristae-specific morphological defect is found in double mutants.

We next assessed planar polarity in the utricle and found comparable polarity in all the genotypes (S6B–S6F Fig). However, we found a significant increase in cristae hair cell misorientation in both *Gpsm2$^{-/-}$* and *Pcdh15$^{n38/n38}$* single mutants relative to controls (Fig 7B, 7C). However, as neither single mutant exhibits circling, these polarity defects alone appear insufficient to explain the phenotype. Notably, hair cell misorientation was further exacerbated in double mutants, although it is uncertain whether this enhanced disorganisation is the sole cause of the circling behaviour. Together, these findings

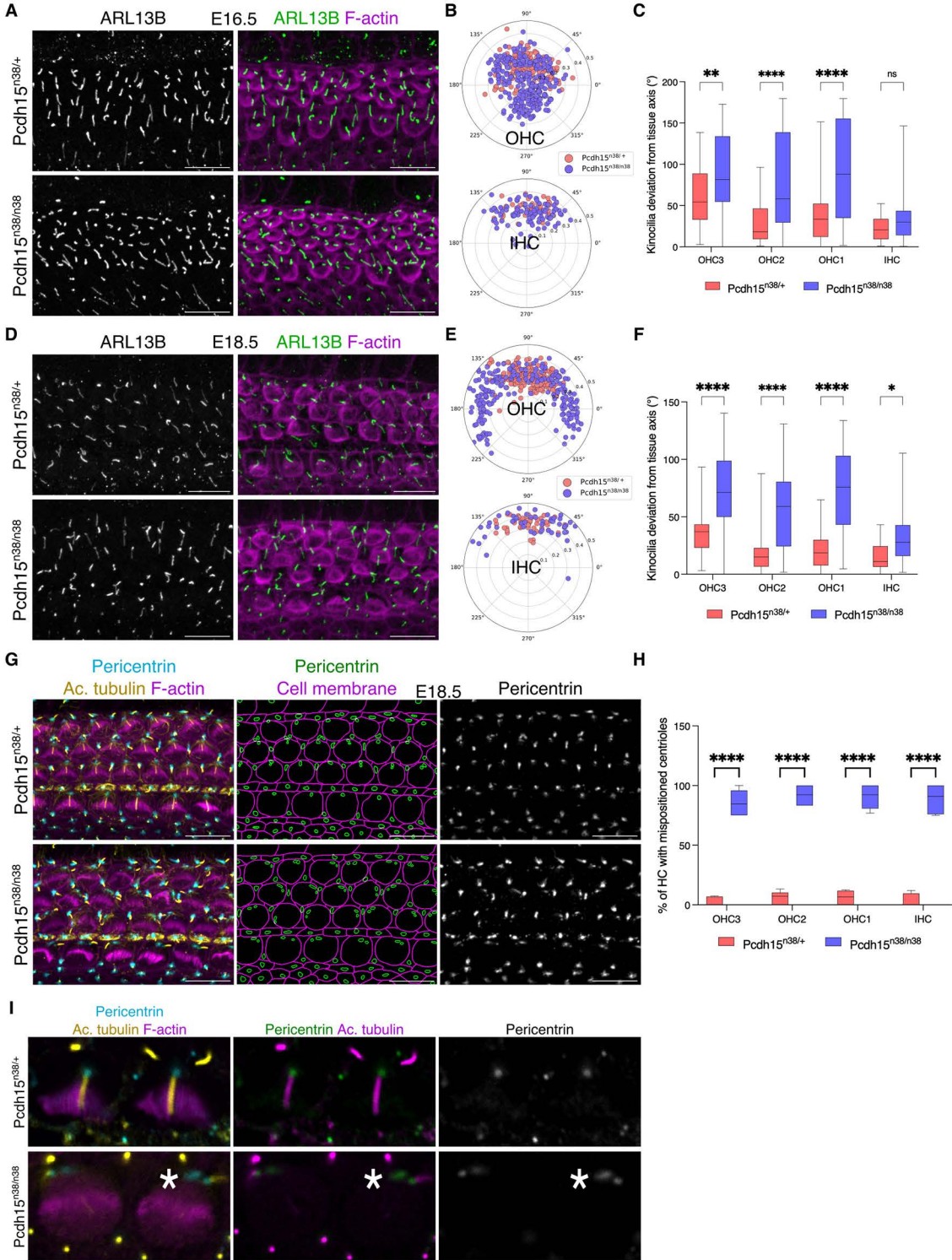

**Fig 4. _Pcdh15_<sup>n38/n38</sup> mice exhibit early hair bundle polarity defects.** (A) Hair bundle polarity is perturbed in _Pcdh15_$^{n38/n38}$ hair cells of E16.5 OC (mid). Cilia are marked with ARL13B (green) and stereocilia with phalloidin (magenta). (Scale bar = 10 μm). (B) Polar projections marking kinocilia positions in E16.5 stage (mid) hair cells show perturbed hair bundle polarity in _Pcdh15_$^{n38/n38}$ (slate blue). (Sample size for OHC/IHC, _Pcdh15_$^{n38/+}$ = 147/48, _Pcdh15_$^{n38/n38}$ = 264/91). (C) Kinocilia are deviated significantly from the tissue axis in _Pcdh15_$^{n38/n38}$ (slate blue) OHC as compared to _Pcdh15_$^{n38/+}$ (pink) (E16.5,

mid). (Two-way Anova with Tukey's multiple comparisons test, ns=P>0.05, *=P<0.05, **=P<0.01, ***=P<0.001, ****=P<0.0001). (Sample size for OHC3/OHC2/OHC1/IHC, $Pcdh15^{n38/+}$=41/53/53/48, $Pcdh15^{n38/n38}$=79/95/90/91). (D) The perturbation in hair bundle polarity continues in $Pcdh15^{n38/n38}$ hair cells of E18.5 OC (mid). Cilia are marked with ARL13B (green) and stereocilia with phalloidin (magenta). (Scale bar=10 μm). (E) Polar projections marking kinocilia positions in E18.5 stage (mid) hair cells show perturbed hair bundle polarity in $Pcdh15^{n38/n38}$ (slate blue). (Sample size for OHC/IHC, $Pcdh15^{n38/+}$=189/51, $Pcdh15^{n38/n38}$=195/50). (F) Kinocilia are deviated significantly from the tissue axis in $Pcdh15^{n38/n38}$ (slate blue) HC (E18.5, mid). (Two-way Anova with Tukey's multiple comparisons test, ns=P>0.05, *=P<0.05, **=P<0.01, ***=P<0.001, ****=P<0.0001). (Sample size for OHC3/OHC2/OHC1/IHC, $Pcdh15^{n38/+}$=60/65/64/51, $Pcdh15^{n38/n38}$=65/65/65/50). (G) Immunostaining and segmentation of the centrioles (marked with pericentrin) show their mispositioning in $Pcdh15^{n38/n38}$ mice hair cells (E18.5, base). (Scale bar=10 μm). (H) Quantitation of the percentage of HC shows centrioles mispositioning in $Pcdh15^{n38/n38}$ mice as compared to the control. (Two-way Anova with Tukey's multiple comparisons test, ns=P>0.05, *=P<0.05, **=P<0.01, ***=P<0.001, ****=P<0.0001). (Sample size for OHC3/OHC2/OHC1/IHC, $Pcdh15^{n38/+}$=87/85/86/79, $Pcdh15^{n38/n38}$=62/60/63/59). (I) A zoomed image of a single optical section of the stained centrioles (marked with pericentrin) connected to the kinocilium (marked with acetylated tubulin) shows the absence of their orthogonal position (asterisk) to each other in IHC of $Pcdh15^{n38/n38}$ mice (E18.5, base).

suggest a cumulative effect of *Pcdh15-CD2* and *GPSM2* loss on vestibular hair cell morphology and polarity, which may contribute to, but do not unequivocally explain, the circling behaviour. Taken together, these data suggest a genetic interaction between *Pcdh15* and *Gpsm2*.

GPSM2 directs the restricted expression of Gαi, which shows a medial expansion in $Pcdh15^{n38/n38}$ (Fig 5A–5C). Gαi interacts with the coiled-coil domain containing proteins, Ccdc88a (Girdin) and Ccdc88c (Daple) [41]. To further understand the nature of this interaction, we investigated the expression of Daple and Girdin in *Pcdh15* mutants. We find that their localisation is in the lateral HC junction closest to the kinocilium, and is unchanged in $Pcdh15^{n38/n38}$ cochlea (Fig 8A). We next asked if the expression of PCDH15 is altered in *Gpsm2*$^{-/-}$ cochlea. We first used an N-terminal specific antibody that recognises all the major isoforms of PCDH15. We found that PCDH15 expression is unperturbed in *Gpsm2*$^{-/-}$ HC (Fig 8B). To ask if the CD2 isoform was affected, we crossed $Pcdh15^{n38YF/n38YF}$ into *Gpsm2*$^{-/-}$ background and assessed the expression using HA-tag. We find that PCDH15-CD2-HA still localises to the tip of stereocilia in these mutants (Fig 8C). These data suggest that *Pcdh15* and *Gpsm2* function together during inner ear development.

### PCDH15-CD2 expression rescues the intrinsic hair bundle polarity defects

Recently, mini PCDH15-CD2 mediated rescue was reported in a *Pcdh15* mutant mouse. Here, the *Pcdh15* allele had been removed using a *Myo15-cre* [42]. Cre is active after intrinsic polarity has already developed in these mice. We asked if re-introducing the *Pcdh15* function could rescue the intrinsic polarity defects observed in $Pcdh15^{n38/n38}$. We reasoned that by controlling the timing of Cre expression, we could induce recombination such that the mutant $Pcdh15^{n38/n38}$ was replaced by the functional $Pcdh15^{n38YF/n38YF}$ allele (Fig 2A, 2B). We used an *Atoh1-CreERT2* line, in which Cre can be activated by tamoxifen induction. We injected dams carrying E17.5 $Pcdh15^{n38/n38}$ mutants with a 5 mg/40 g dose of tamoxifen and determined the resulting phenotype at E19.5 (Fig 9A).

In $Pcdh15^{n38/n38}$ littermate controls, all HC showed kinocilia decoupling from stereocilia, and HC showed intrinsic polarity defects (Fig 9B–9D). These cochleae did not express PCDH15-CD2-HA (Fig 9B, 9C). In *Atoh1-CreET2*$^{+/-}$::$Pcdh15^{n38/n38}$ animals, while *Atoh1-CreERT2* induction failed to induce recombination in IHC, we observed mosaic expression of PCDH15-CD2-HA in OHC (Fig 9B, 9C). In HA-positive HC, we found that kinocilia and stereocilia were coupled (Fig 9B). We quantified the kinocilium position in hair cells, differentiating whether they expressed PCDH15-CD2-HA. We found that the kinocilia of PCDH15-CD2-HA expressing OHC align with the tissue axis (26˚±30˚), in contrast to the non-expressing OHC (48˚±37˚) (Fig 9D).

### Discussion

Our study identifies a role for PCDH15-CD2 in regulating intrinsic hair cell polarity, specifically through the coordination of basal body positioning and Gαi localisation. While PCDH15 is classically defined as a component of kinocilial and stereocilial links and is essential for mechanoelectrical transduction, our findings reveal that the CD2 isoform contributes earlier

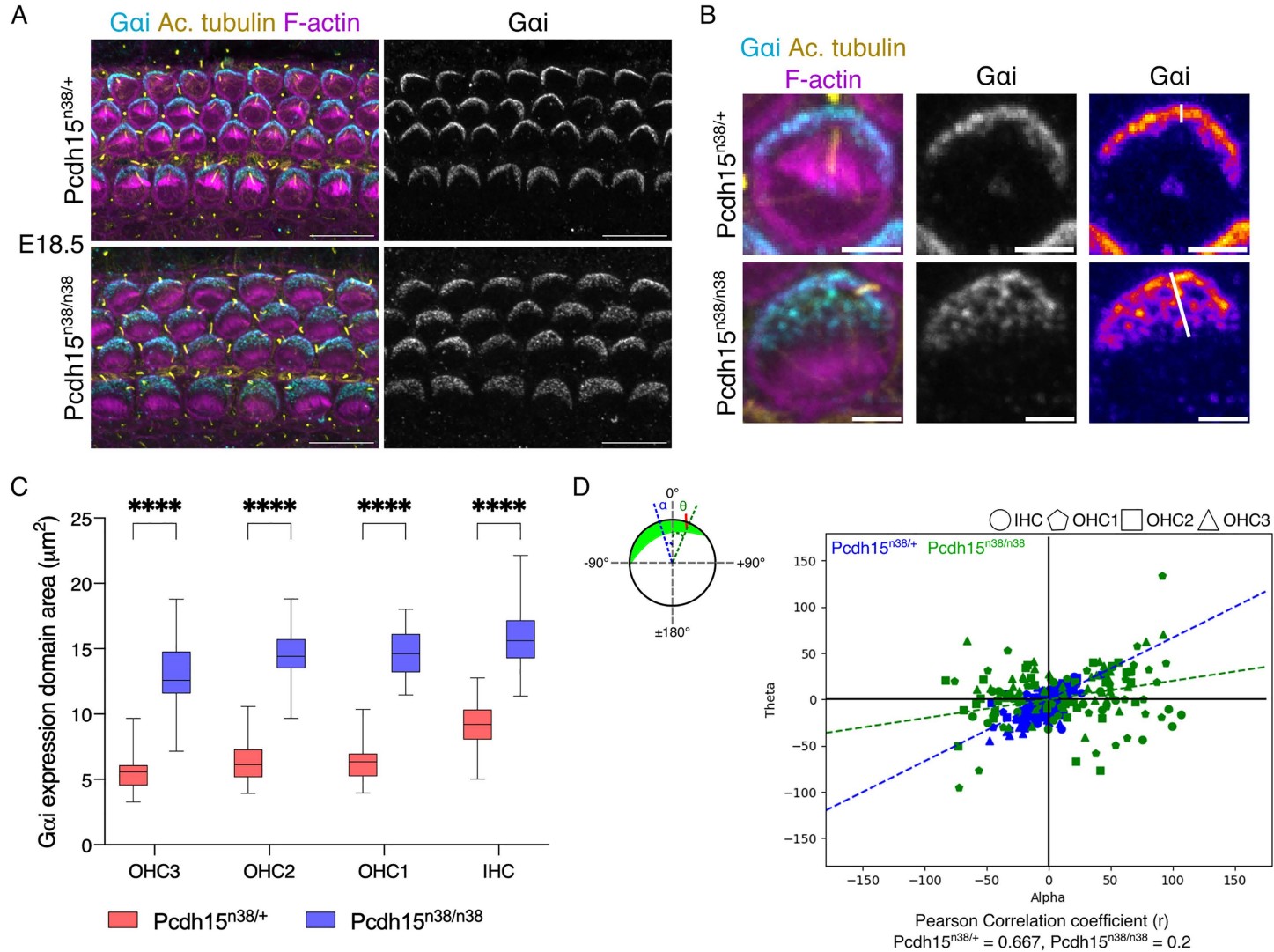

**Fig 5. Kinocilium position is not coordinated with G α i expression in *Pcdh15^{n38/n38}* mice hair cells.** (A) Gαi expression (cyan) is perturbed with respect to the kinocilium position (yellow, marked with acetylated tubulin) in E18.5 (base) hair cells of *Pcdh15^{n38/n38}* mice. (Scale bar = 10 μm). (B) Gαi expression domain (cyan) is extended medially in HC of *Pcdh15^{n38/n38}* mice. (Scale bar = 2 μm). (C) In comparison to *Pcdh15^{n38/+}* mice, the Gαi expression domain is spread significantly in HC of *Pcdh15^{n38/n38}* mice (E18.5, base). (Two-way Anova with Tukey's multiple comparisons test, ns = P > 0.05, * = P < 0.05, ** = P < 0.01, *** = P < 0.001, **** = P < 0.0001). (Sample size for OHC3/OHC2/OHC1/IHC, *Pcdh15^{n38/+}* = 51/54/50/46, *Pcdh15^{n38/n38}* = 55/57/56/53). (D) Correlation between the kinocilium position and Gαi expression domain is reduced in HC of *Pcdh15^{n38/n38}* mice (green) when compared to *Pcdh15^{n38/+}* (blue) (E18.5, base). Alpha (α) is the angle measured for the kinocilium with the tissue axis, and theta (θ) is the angle measured for the Gαi expression domain (mid) with the tissue axis. (Sample size for OHC3/OHC2/OHC1/IHC, *Pcdh15^{n38/+}* = 51/54/50/46, *Pcdh15^{n38/n38}* = 55/57/56/53).

to the apical architecture of hair cells. These functions appear to be independent of tyrosine phosphorylation, as the phospho-mutant allele (*n38YF*) fully rescues polarity defects.

Hair cells display two distinct but interconnected types of polarity: intrinsic polarity, which defines the spatial organisation of subcellular components such as the kinocilium, G-protein domains, and actin-based stereocilia within a single cell, and planar cell polarity (PCP), which aligns this intrinsic polarity across the epithelial sheet [11]. While PCP components such as *Vangl2* and *Celsr1* coordinate tissue-level alignment, intrinsic polarity mechanisms must integrate cytoskeletal,

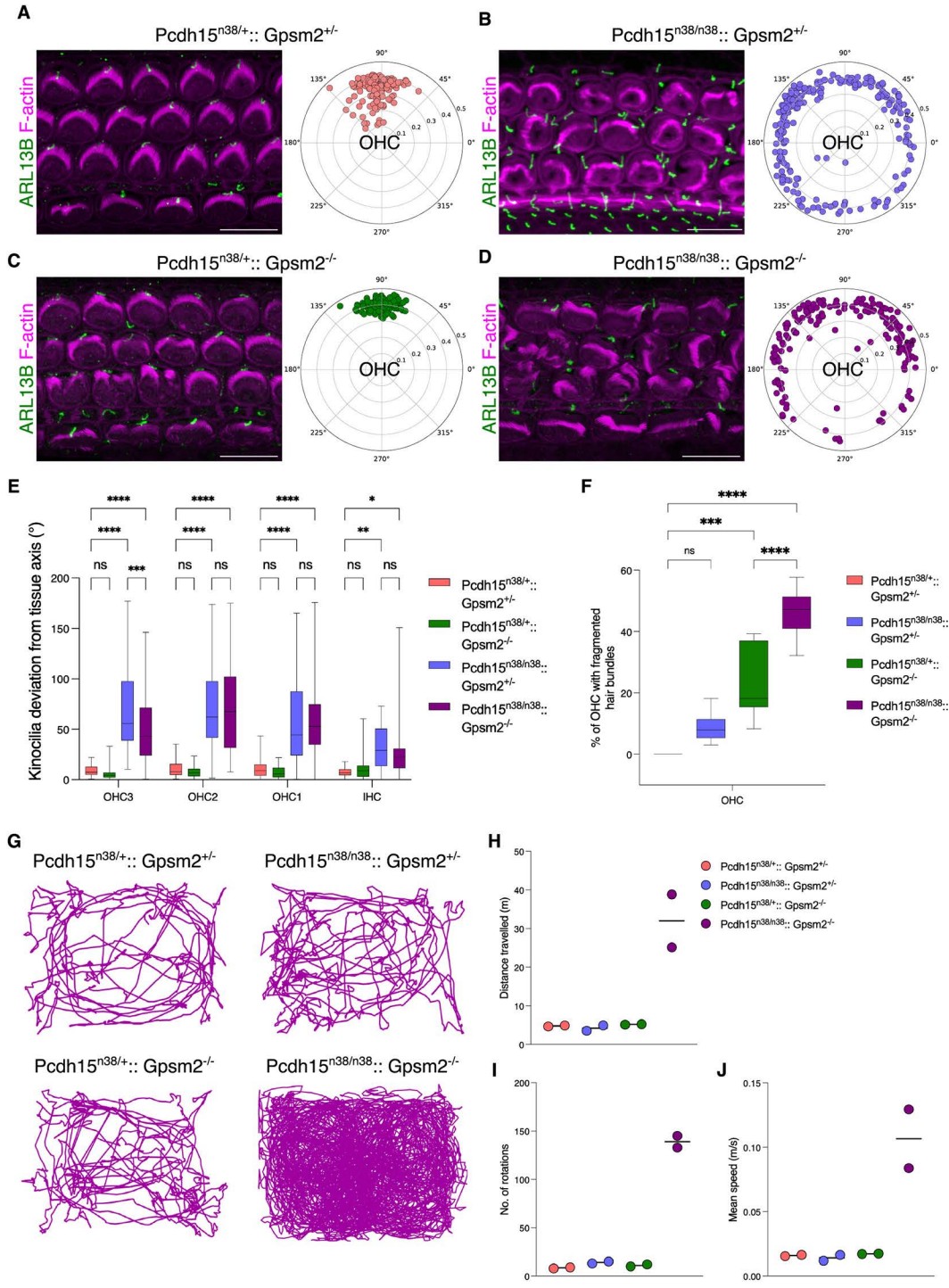

**Fig 6. *Pcdh15-CD2* and *Gpsm2* show genetic interaction during hair bundle development.** (A-D) Whole mount OC preparation from mice of the indicated genotype at P1 (base). Cilia are marked with ARL13B (green) and stereocilia with phalloidin (magenta). In comparison to *Pcdh15^{n38/+}:: Gpsm2^{+/-}* (A) (pink), the hair bundle polarity of P1 OHC is perturbed in *Pcdh15^{n38/n38}:: Gpsm2^{+/-}* OC (B) (slate blue) and *Pcdh15^{n38/n38}:: Gpsm2^{-/-}* (D) (dark magenta) OC. Hair bundle polarity is normal in *Pcdh15^{n38/+}:: Gpsm2^{-/-}* (green) OC (C) (Scale bar = 10 µm). (Sample size for OHC/IHC, *Pcdh15^{n38/+}:: Gpsm2^{+/-}* = 120/36, *Pcdh15^{n38/+}:: Gpsm2^{-/-}* = 156/42, *Pcdh15^{n38/n38}:: Gpsm2^{+/-}* = 180/24, *Pcdh15^{n38/n38}:: Gpsm2^{-/-}* = 174/54). (E) Kinocilia are deviated significantly from the tissue axis in HC of *Pcdh15^{n38/n38}:: Gpsm2^{+/-}* (slate blue) & *Pcdh15^{n38/n38}:: Gpsm2^{-/-}* (dark magenta) and aligned along the tissue axis in *Pcdh15^{n38/+}:: Gpsm2^{+/-}* (pink) & *Pcdh15^{n38/+}:: Gpsm2^{-/-}* (green) OC (P1, base). (Two-way Anova with Tukey's multiple comparisons test, ns = P > 0.05,

*=$P<0.05$, **=$P<0.01$, ***=$P<0.001$, ****=$P<0.0001$). (Sample size for OHC3/OHC2/OHC1/IHC, $Pcdh15^{n38/+}$:: $Gpsm2^{+/-}=40/40/40/36$, $Pcdh15^{n38/+}$:: $Gpsm2^{-/-}=50/53/53/42$, $Pcdh15^{n38/n38}$:: $Gpsm2^{+/-}=60/59/61/24$, $Pcdh15^{n38/n38}$:: $Gpsm2^{-/-}=57/59/58/54$). (F) Hair bundles are fragmented in $Pcdh15^{n38/+}$:: $Gpsm2^{-/-}$ (green) OC which is further exacerbated in $Pcdh15^{n38/n38}$:: $Gpsm2^{-/-}$ (dark magenta) (P1, base). (Ordinary one-way Anova with Tukey's multiple comparisons test, ns=$P>0.05$, *=$P<0.05$, **=$P<0.01$, ***=$P<0.001$, ****=$P<0.0001$). (Sample size for OHC, $Pcdh15^{n38/+}$:: $Gpsm2^{+/-}=157$, $Pcdh15^{n38/+}$:: $Gpsm2^{-/-}=156$, $Pcdh15^{n38/n38}$:: $Gpsm2^{+/-}=223$, $Pcdh15^{n38/n38}$:: $Gpsm2^{-/-}=174$). (G) Circling activity is observed only in $Pcdh15^{n38/n38}$:: $Gpsm2^{-/-}$ mice and not in $Pcdh15^{n38/+}$:: $Gpsm2^{+/-}$ or $Pcdh15^{n38/+}$:: $Gpsm2^{-/-}$ mice as compared to control mice when observed under open-field test (OFT) for 5 min. (H-J) $Pcdh15^{n38/n38}$:: $Gpsm2^{-/-}$ mice cover more distance, move faster and rotate more than control animals (each circle is an individual animal).

centrosomal, and membrane-associated cues to generate and maintain internal asymmetry [12]. Our data demonstrate that *Pcdh15-CD2* mutants exhibit defects in intrinsic polarity, particularly in the positioning of the kinocilium relative to the Gαi domain. These defects in intrinsic polarity are similar to the defects observed in the Ames Waltzer, *Pcdh15* mutant mouse [23,43].

The function of *Pcdh15-CD2* in intrinsic polarity has been ascribed to the absence of kinocilial links. This is supported by the intrinsic polarity defects observed in mutations in the binding partner of *Pcdh15*, *Cdh23* [23]. However, examining the timing of kinocilium migration and the expression of PCDH15-CD2 may suggest an earlier role in this process. Kinocilial links are only observed once stereocilia begin to elongate, and are first visible in the OHC found at the base of the cochlea at E17.5 [30]. This is after the initial centrifugal relocalisation of the kinocilia [8–10]. Indeed, intrinsic polarity defects in *Pcdh15-n38* mutants are already evident by E16.5, prior to the appearance of kinocilial links. Furthermore, using an HA-tagged PCDH15-CD2 allele, we detected expression of the CD2 isoform at E15.5-E16.5, localised around the base of the kinocilium. This suggests a possible earlier role for *Pcdh15-CD2* independent of its role in kinocilial link formation.

In wild-type cochlear hair cells, the kinocilium relocates from a central to peripheral position in a two-step process: it first moves peripherally and then relocates radially to align with the centre of the Gαi crescent at the apical surface [9,10]. In *Pcdh15-n38* mutants, the kinocilium fails to localise to the centre of the GPSM2-Gαi crescent. This process occurs between E16.5 and E17.5 [9]. It is unlikely that this positioning is initiated through kinocilium–stereocilium adhesion, as these structures remain physically distant at this stage. Furthermore, in our rescue experiment, it is difficult to envision how the kinocilium would recouple with the stereocilia through adhesion alone.

Although the initial steps in intrinsic polarity generation are independent of kinocilial links, further steps likely require functional attachments between PCDH15 in the kinocilium and CDH23 in the tallest stereocilium. These links are likely to be established once the kinocilium is at the centre of the Gαi crescent. The expansion of the bare zone shifts the hair bundle medially [9,10]. In *Pcdh15-n38* mutants, these links are no longer present, and the kinocilium is not anchored to the stereociliary bundle. The failure in the coordination between the kinocilium and the stereocilia is reflected in the differences between the variation of alignment of kinocilia and that of stereocilia, to the tissue axis in mutants.

In *Pcdh15-n38* mutants, we observed a medial expansion of both GPSM2 and Gαi. This expansion has also been reported in basal body and ciliary mutants, including *Mkks*, *Ift88*, and *Bbs8* [10,14,18]. This suggests that PCDH15-CD2 may function upstream in organising the pericentriolar architecture necessary for correct apical patterning. Support for this comes from the mispositioning of centrioles observed in *Pcdh15-n38* mutants. Such positioning has also been observed in *Kif3a* mutants. Normally, the mother centriole (kinocilium basal body) is apically anchored, while the daughter centriole lies more basally. In *Kif3a* mutants, that difference in apicobasal position is lost [19]. Perturbed RAC-PAK signalling is thought to underlie the perturbed basal body position in the *Kif3a* mutant, with active RAC1 recruited to the pericentriolar matrix through the microtubule-associated protein, LIS1 [44]. LIS1 is also required for the dynein activity and localisation around the kinocilium centrosome [45,46]. Our data suggest that PCDH15-CD2 is also localised around the basal body, suggesting a possible link through which CD2 and the RAC-PAK pathway could influence basal body position and apical domain organisation. However, how exactly this could impact the localisation of Gαi3 and GPSM2 is unclear.

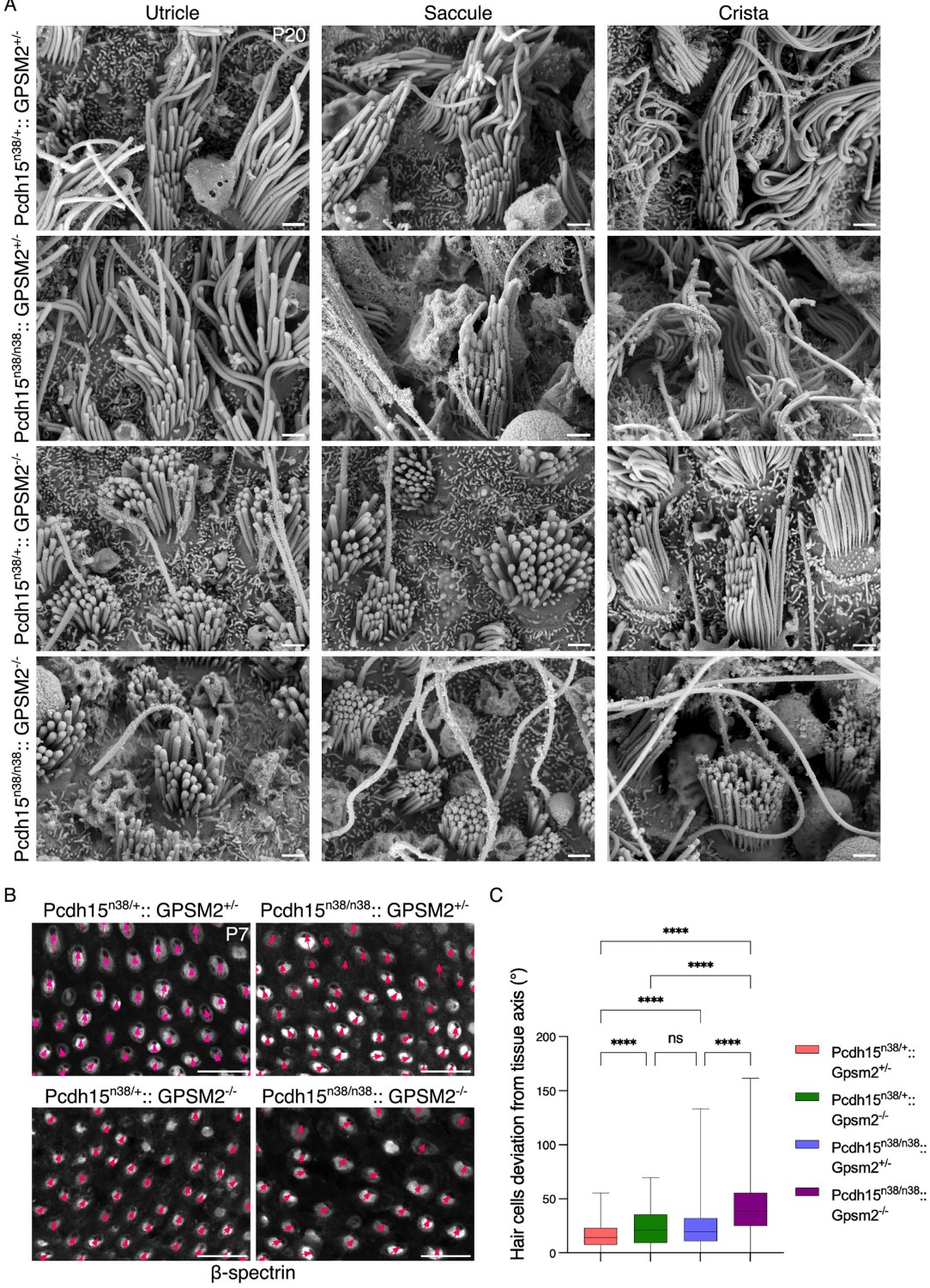

**Fig 7. *Pcdh15-CD2* and *Gpsm2* show genetic interaction during hair bundle development of the vestibular system.** (A) SEM imaging shows smaller stereocilia bundles in the hair cells of all three vestibular structures (utricle, saccule and crista) of *Pcdh15^(n38/+):: Gpsm2^(-/-)* & *Pcdh15^(n38/n38)::
Gpsm2^(-/-)* genotypes, but not in *Pcdh15^(n38/n38):: Gpsm2^(+/-)*. The stereocilia gradation is preserved to some extent in the crista of *Pcdh15^(n38/+):: Gpsm2^(-/-)* mice, but is lost in *Pcdh15^(n38/n38):: Gpsm2^(-/-)* (P20, Scale bar = 1 μm). (B - C) Direction of hair cells in the cristae (marked with red arrows) shows a significant

increase in their misorientation in *Pcdh15^{n38/n38}:: Gpsm2^{+/-}* and *Pcdh15^{n38/+}:: Gpsm2^{-/-}* as compared to *Pcdh15^{n38/+}:: Gpsm2^{+/-}*, which is further increased in the *Pcdh15^{n38/n38}:: Gpsm2^{-/-}* mice. (P7, Scale bar = 20 μm). (Ordinary one-way Anova with Tukey's multiple comparisons test, ns = P > 0.05, * = P < 0.05, ** = P < 0.01, *** = P < 0.001, **** = P < 0.0001). (Sample size for HC, *Pcdh15^{n38/+}:: Gpsm2^{+/-}* = 341, *Pcdh15^{n38/+}:: Gpsm2^{-/-}* = 350, *Pcdh15^{n38/n38}:: Gpsm2^{+/-}* = 356, *Pcdh15^{n38/n38}:: Gpsm2^{-/-}* = 354).

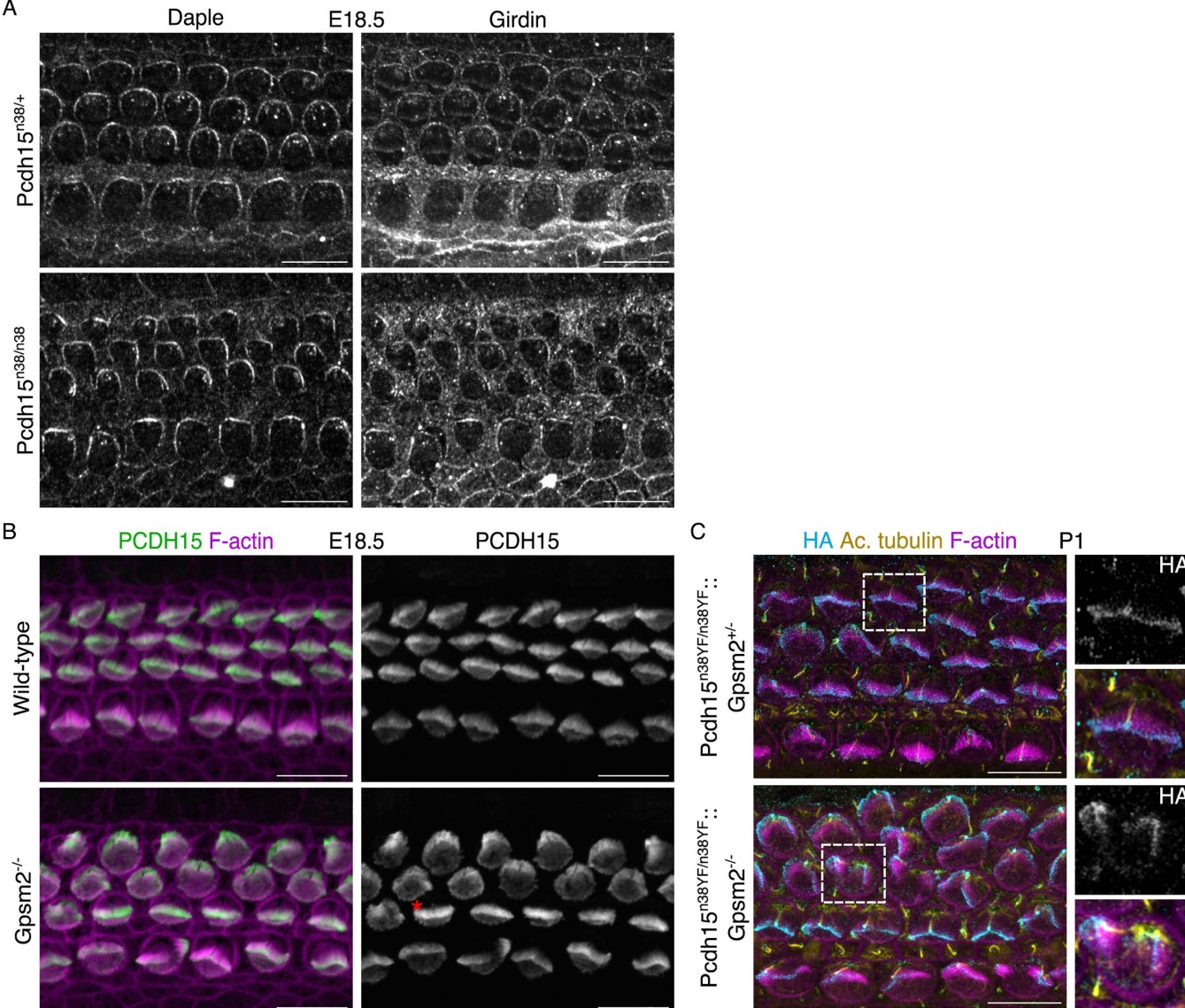

**Fig 8. PCDH15-CD2 and GPSM2 proteins operate in parallel pathways during hair bundle development.** (A) Daple & Girdin proteins (grey) are localised on the neural edge of the HC of both *Pcdh15^{n38/+}* and *Pcdh15^{n38/n38}* mice (E18.5, base). (Scale bar = 10 μm). (B) PCDH15 protein (green) is localised on the hair bundles (even in fragmented hair bundles, marked with red asterisk) of *Gpsm2* mutant HC (E18.5, mid). (Scale bar = 10 μm). (C) PCDH15-CD2-HA (cyan) is localised on both kinocilium (yellow, marked with acetylated tubulin) and stereocilia (magenta, marked with phalloidin) in *Gpsm2* mutant HC (P1, base). (Scale bar = 10 μm).

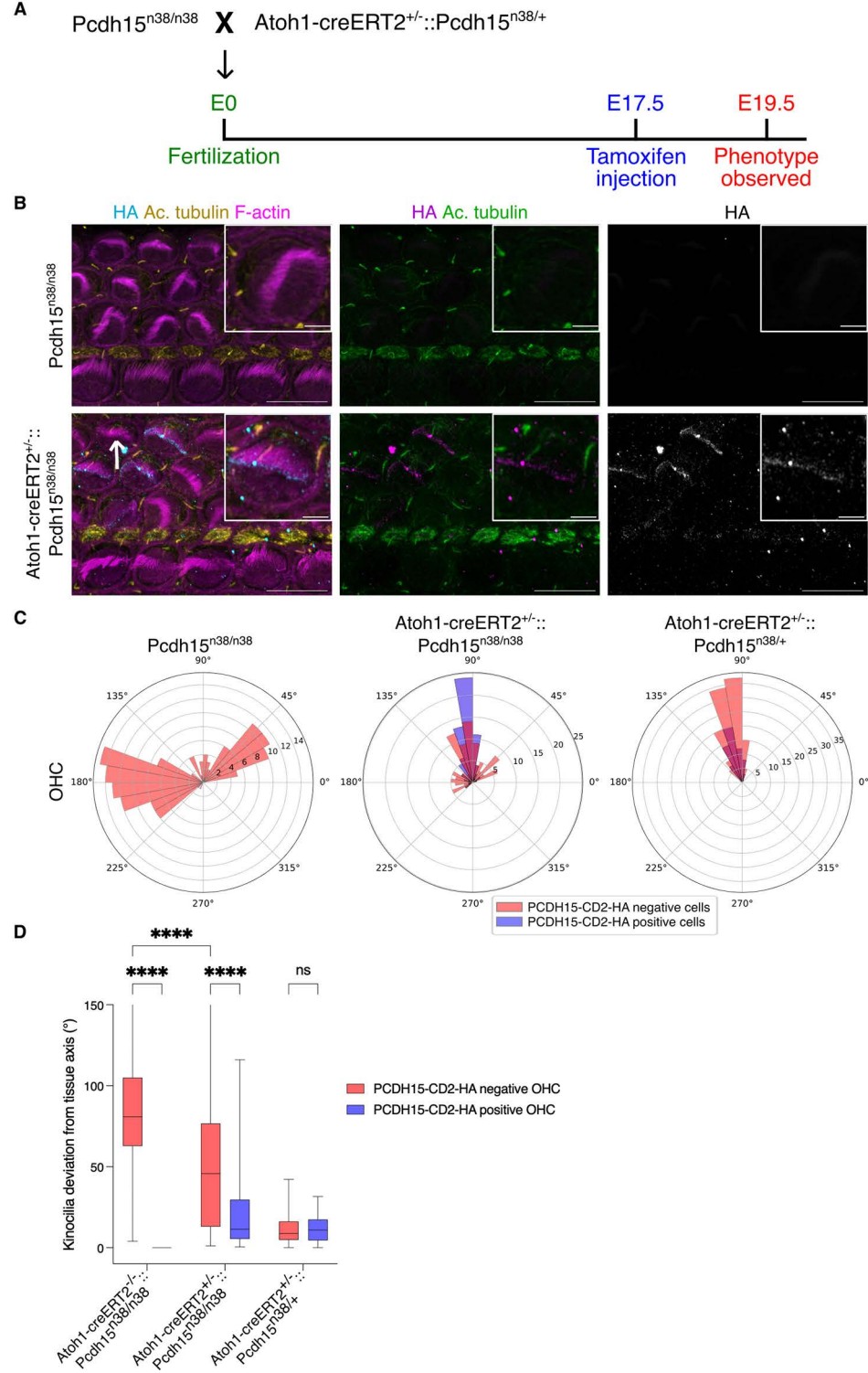

**Fig 9. Re-initiation of PCDH15-CD2 expression rescues the intrinsic hair bundle polarity defects.** (A) Schematic shows the experimental scheme for the rescue of PCDH15-CD2 expression. (B) PCDH15-CD2-HA (cyan) expression is rescued upon *Atoh1-creERT2* induction. Kinocilium (yellow, marked with acetylated tubulin) is associated with stereocilia (magenta, marked with phalloidin), specifically in HC with PCDH15-CD2-HA expression. Few HC showed kinocilia attached to the stereocilia bundles without PCDH15-CD2-HA expression as well (marked with white arrow). (E19.5, base,

Scale bar = 10 μm). (C) Circular histogram shows rescue in hair bundle polarity upon Atoh1creERT2-mediated induction of PCDH15-CD2-HA (slate blue) (E19.5, base). (D) Kinocilia deviation from the tissue axis is improved in HC with PCDH15-CD2-HA expression (slate blue) (E19.5, base). (Two-way Anova with Tukey's multiple comparisons test, ns = P > 0.05, * = P < 0.05, ** = P < 0.01, *** = P < 0.001, **** = P < 0.0001). (Sample size for PCDH15-CD2-HA negative OHC/PCDH15-CD2-HA positive OHC, $Pcdh15^{n38n38}$ = 191/0, $Atoh1cre\text{-}ERT2^{+/-}:: Pcdh15^{n38/n38}$ = 121/89, $Atoh1cre\text{-}ERT2^{+/-}:: Pcdh15^{n38/+}$ = 127/66).

Although distinct molecular pathways specify kinocilium and Gαi domains, their spatial convergence appears necessary for proper bundle morphogenesis. This loss of coordination can affect the cohesion of the stereociliary bundle. This is supported by our observation that *Pcdh15-n38:: Gpsm2* double mutants exhibit exacerbated bundle fragmentation. These phenotypes parallel those observed in *Daple* mutants, where kinocilium-Gαi misalignment results in disorganised bundles [41]. Thus, intrinsic polarity defects in basal body and Gαi alignment appear to drive secondary consequences in bundle assembly. Moreover, the position of kinocilia can influence planar polarity. *Bbs8* is known to influence the trafficking of VANGL2, and overall epithelial organisation may be affected as a result [18]. However, in our analysis of *Pcdh15-n38* mutants, we did not detect an alteration in VANGL2 localisation. Recent data suggest that the kinocilium's position alters junctional mechanics during chick cochlea extension, altering polarity [47]. *Pcdh15-CD2* may change the ability of the kinocilia to influence neighbouring cellular dynamics, leading to defects in planar polarity. The restoration of PCDH15-CD2 expression via Cre-mediated recombination partially rescued intrinsic polarity, confirming that PCDH15-CD2 is likely sufficient to align the Gαi domains. This result underscores the conclusion that PCDH15-CD2 is a critical mediator of intrinsic polarity in developing hair cells.

Interestingly, although *Pcdh15-n38* and *Gpsm2* single mutants exhibit hearing loss, only the double mutants demonstrate vestibular-related behaviours such as circling and hyperactivity. One reason could be the shorter stereociliary bundles observed in double *Pcdh15-n38::Gpsm2* mutants. However, *Gpsm2* single mutants also show shorter stereocilia, but do not show a circling behaviour. This is in contrast to earlier analysis of the *mPins* allele of *Gpsm2* [48]. Here, a vestibular phenotype was observed in the single mutant, and utricular hair cells show reduced stereocilia length. The reason for the discrepancy between the two alleles is unclear. However, the *mPins* allele may produce a truncated protein, which could act as a dominant negative. Double mutants show an exacerbated intrinsic polarity defect and also show reduced gradation in the stereociliary bundle of the cristae. Together with a reduction in kinociliary links, these may be sufficient to impair vestibular function.

Taken together, our findings suggest PCDH15-CD2 integrates cytoskeletal, ciliary, and membrane-associated signals to orchestrate intrinsic polarity in sensory hair cells. While planar polarity ensures tissue-wide alignment of cellular orientation, intrinsic polarity mechanisms, anchored by basal body organisation and Gαi localisation, set the stage for this higher-order coordination. Understanding how these two layers of polarity interact will be essential for decoding morphogenetic control in sensory epithelia.

## Supporting information

**S1 Fig.** ***Pcdh15*<sup>n</sup>38<sup>/n</sup>38 mice are functionally null and act similar to *Pcdh15-ΔCD2* mutants.** (A) A small fragment of *Pcdh15* exon 38 was amplified in WT (167 bp) and $Pcdh15^{n38/n38}$ (247 bp) animals for genotyping. (B) Full-length western blot of Fig 1D. (C) mRNA level for CD2 domain is normal in $Pcdh15^{n38/n38}$ cochlea. (D) VANGL2 protein (green) localisation (highlighted with red arrows) is unperturbed in $Pcdh15^{n38/n38}$ OC (E17.5, base). (Scale bar = 10 μm). (E) Stereocilia bundles are aligned normally with the tissue axis in $Pcdh15^{n38/+}$ (pink), but significantly deviated in $Pcdh15^{n38/n38}$ (slate blue) (P0, mid). (Two-way Anova with Tukey's multiple comparisons test, ns = P > 0.05, * = P < 0.05, ** = P < 0.01, *** = P < 0.001, **** = P < 0.0001). (Sample size for OHC3/OHC2/OHC1, $Pcdh15^{n38/+}$ = 69/67/72, $Pcdh15^{n38/n38}$ = 69/68/69). (TIF)

**S2 Fig. Tyrosine phosphorylation of the CD2 domain is not essential for PCDH15 function.** (A) Part of WT and *Pcdh15^{n38YF/n38YF}* exon 38 are amplified in different sizes (167 bp and 207 bp, respectively) with the same primer pair for genotyping. (B) PCDH15-CD2-HA protein (green) is colocalised with PCDH15 (cyan) in the hair bundles (P0, mid). (Scale bar = 10 μm). (C) PCDH15-CD2-HA protein (green) is localised at the tip of stereocilia (magenta, marked with phalloidin) at E18.5 (mid). At P5 (mid), this expression is localised in all three rows of stereocilia and is refined more in P15 stereocilia (mid). (Scale bar = 10 μm). (D) Full-length western blot of Fig 2E. (E) SEM imaging shows normal hair bundle polarity in both *Pcdh15^{n38/+}* and *Pcdh15^{n38YF/n38YF}* HC (P0, mid). (Scale bar = 5 μm). (F) MET channels are open in HC of *Pcdh15^{n38/n38}* and *Pcdh15^{n38YF/n38YF}* mice at P5. FM1–43 FX dye (green) is used to detect open MET channels. (Scale bar = 10 μm). (G) Positive Preyer's reflex response measurement shows no response in *Pcdh15^{n38/n38}* adult animals as compared to the comparable response in wild-type and *Pcdh15^{n38YF/n38YF}* mice. (H-I) PCDH15-CD2-HA protein signal (green) is not present in wild-type mice organ of Corti (H) and utricle & cristae (I) at E16.5 stage (Staining controls). (Scale bar = 10 μm). (TIF)

**S3 Fig. *Pcdh15^{n38/n38}* mice exhibit early hair bundle polarity defects.** (A) A single optical section of the stained centrioles (marked with pericentrin) connected to the kinocilium (marked with acetylated tubulin) shows the absence of their orthogonal position to each other in HC of *Pcdh15^{n38/n38}* mice (E18.5, base). (TIF)

**S4 Fig. GPSM2 expression is perturbed in *Pcdh15^{n38/n38}* mice hair cells.** (A-B) GPSM2 expression domain (green) is extended medially in HC of *Pcdh15^{n38/n38}* mice (E18.5, base). (Scale bar = 10 μm & 2 μm). (C) In comparison to *Pcdh15^{n38/+}* mice, GPSM2 expression domain is spread significantly in HC of *Pcdh15^{n38/n38}* mice (E18.5, base). (Two-way Anova with Tukey's multiple comparisons test, ns = P > 0.05, * = P < 0.05, ** = P < 0.01, *** = P < 0.001, **** = P < 0.0001). (Sample size for OHC3/OHC2/OHC1/IHC, *Pcdh15^{n38/+}* = 59/56/57/52, *Pcdh15^{n38/n38}* = 69/70/66/66). (TIF)

**S5 Fig. *Pcdh15-CD2* and *Gpsm2* show genetic interaction during hair bundle development.** (A) In comparison to *Pcdh15^{n38/+}:: Gpsm2^{+/-}* (pink), the hair bundle polarity (polar projections) is perturbed in P1 IHC of *Pcdh15^{n38/n38}:: Gpsm2^{+/-}* (slate blue) and *Pcdh15^{n38/n38}:: Gpsm2^{-/-}* (dark magenta) OC (base). Hair bundle polarity is normal in *Pcdh15^{n38/+}:: Gpsm2^{-/-}* (green) OC. (B) Hair bundles are fragmented in IHC of only *Pcdh15^{n38/n38}:: Gpsm2^{-/-}* (dark magenta). (Ordinary one-way Anova with Tukey's multiple comparisons test, ns = P > 0.05, * = P < 0.05, ** = P < 0.01, *** = P < 0.001, **** = P < 0.0001). (Sample size for IHC, *Pcdh15^{n38/+}:: Gpsm2^{+/-}* = 36, *Pcdh15^{n38/+}:: Gpsm2^{-/-}* = 42, *Pcdh15^{n38/n38}:: Gpsm2^{+/-}* = 24, *Pcdh15^{n38/n38}:: Gpsm2^{-/-}* = 54). (TIF)

**S6 Fig. *Pcdh15-CD2* and *Gpsm2* show genetic interaction during hair bundle development of the vestibular system.** (A) F-actin (grey) shows smaller stereocilia bundles in the hair cells crista of *Pcdh15^{n38/+}:: Gpsm2^{-/-}*, which is further reduced in *Pcdh15^{n38/n38}:: Gpsm2^{-/-}* mice, but not in *Pcdh15^{n38/n38}:: Gpsm2^{+/-}*. (P7, Scale bar = 20 μm). (B-F) Direction of hair cells in the (D) lateral, (E) LPR, and (F) medial region of the utricle shows comparable orientation in *Pcdh15^{n38/n38}:: Gpsm2^{+/-}*, *Pcdh15^{n38/+}:: Gpsm2^{-/-}* and *Pcdh15^{n38/n38}:: Gpsm2^{-/-}* mice as compared to littermate controls. (P7, Scale bar = 50 & 20 μm). (TIF)

**S1 Movie. *Pcdh15-CD2* and *Gpsm2* show genetic interaction during the hair bundle development of the vestibular system.** Circling activity is observed only in *Pcdh15^{n38/n38}:: Gpsm2^{-/-}* mice and not in *Pcdh15^{n38/+}:: Gpsm2^{+/-}* or *Pcdh15^{n38/+}:: Gpsm2^{-/-}* mice as compared to control mice when observed under the open-field test (OFT). (MP4)

**S1 Table.  Genotyping and RT-PCR conditions used for different genes.** These include their specific primers, annealing temperature, and amplified product size.
(DOCX)

**S2 Table.  List of primary antibodies used for immunofluorescence assays and western blotting.**
(DOCX)

**S3 Table.  List of secondary antibodies used for immunofluorescence assays.**
(DOCX)

**S1 Data.  The raw data of all the measurements and analyses used in this manuscript are provided as an Excel sheet under the file name Pcdh15 paper raw data.xlsx.**
(XLSX)

**S1 Script.  A Python script for measuring alpha-theta correlation is provided under the file name analyse.py.**
(PY)

## Acknowledgments

We acknowledge the support of the Animal Care and Resources Centre, the Electron Microscopy Facility, and the Central Imaging Facility at NCBS. We thank Prof. Fumio Matsuzaki for providing the GPSM2 antibody and *Gpsm2* mutant mice. We thank Prof. Anna Lysakowski for suggestions on the sample preparation of adult mouse vestibular structures for SEM. We thank Hrishikesh Nambisan and Arkadeep Bhattacharjee for help with the mouse behaviour analysis and Digvijay Lalwani for data analysis. We thank members of the Ear Lab at NCBS for their feedback and discussions.

## Author contributions

**Conceptualization:** Raman Kaushik, Raj K Ladher.

**Data curation:** Raj K Ladher.

**Formal analysis:** Anubhav Prakash, Raj K Ladher.

**Funding acquisition:** Raj K Ladher.

**Investigation:** Raman Kaushik, Shivangi Pandey, Anubhav Prakash, Fenil Ganatra.

**Methodology:** Raman Kaushik, Shivangi Pandey, Fenil Ganatra, Takaya Abe, Hiroshi Kiyonari.

**Project administration:** Raj K Ladher.

**Resources:** Takaya Abe, Hiroshi Kiyonari.

**Supervision:** Raj K Ladher.

**Writing – original draft:** Raman Kaushik, Raj K Ladher.

**Writing – review & editing:** Raman Kaushik, Shivangi Pandey, Anubhav Prakash, Takaya Abe, Raj K Ladher.

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
