## [Decision Letter · Decision Letter 0]

10 Mar 2025

PGENETICS-D-24-01537

Role Of Pcdh15 In The Development Of Intrinsic Polarity Of Inner Ear Hair Cells

PLOS Genetics

Dear Dr. Ladher,

Thank you for submitting your manuscript to PLOS Genetics. After careful consideration, we feel that it has merit but does not fully meet PLOS Genetics's publication criteria as it currently stands. Therefore, we invite you to submit a revised version of the manuscript that addresses the points raised during the review process.

Please submit your revised manuscript within 60 days May 09 2025 11:59PM. If you will need more time than this to complete your revisions, please reply to this message or contact the journal office at plosgenetics@plos.org. Please include the following items when submitting your revised manuscript:

We look forward to receiving your revised manuscript.

Kind regards,

Nandan Nerurkar

Academic Editor

PLOS Genetics

Fengwei Yu

Section Editor

PLOS Genetics

Aimée Dudley

Editor-in-Chief

PLOS Genetics

Anne Goriely

Editor-in-Chief

PLOS Genetics

**Journal Requirements:**

https://journals.plos.org/plosgenetics/s/submission-guidelines#loc-parts-of-a-submission

3) We noticed that you used the phrase 'data not shown' in the manuscript. We do not allow these references, as the PLOS data access policy requires that all data be either published with the manuscript or made available in a publicly accessible database. Please amend the supplementary material to include the referenced data or remove the references.

4) We do not publish any copyright or trademark symbols that usually accompany proprietary names, eg ©,  ®, or TM  (e.g. next to drug or reagent names). Therefore please remove all instances of trademark/copyright symbols throughout the text, including:

- TM on pages: 7, and 37.

5) Thank you for including an Ethics Statement for your study. Please include:

i) The full name(s) of the Institutional Review Board(s) or Ethics Committee(s)

ii) The approval number(s), or a statement that approval was granted by the named board(s).

6) Please upload all main figures as separate Figure files in .tif or .eps format. For more information about how to convert and format your figure files please see our guidelines: 

7) Please upload a copy of Fig. S4A which you refer to in your text on page 16. Or, if the figure is no longer to be included as part of the submission please remove all reference to it within the text.

8) We have noticed that you have uploaded Supporting Information files, but you have not included a complete list of legends. Please add a full list of legends for your Supporting Information file (Supplementary Movie )after the references list.

9) We notice that your supplementary Tables are included in the manuscript file. Please remove them and upload them with the file type 'Supporting Information'. Please ensure that each Supporting Information file has a legend listed in the manuscript after the references list.

10) Some material included in your submission may be copyrighted. According to PLOSu2019s copyright policy, authors who use figures or other material (e.g., graphics, clipart, maps) from another author or copyright holder must demonstrate or obtain permission to publish this material under the Creative Commons Attribution 4.0 International (CC BY 4.0) License used by PLOS journals. Please closely review the details of PLOSu2019s copyright requirements here: PLOS Licenses and Copyright. If you need to request permissions from a copyright holder, you may use PLOS's Copyright Content Permission form.

Potential Copyright Issues:

i) Please confirm (a) that you are the photographer of Supplementary Movie, or (b) provide written permission from the photographer to publish it under our CC BY 4.0 license.

ii) Figure 2A. Please confirm whether you drew the images / clip-art within the figure panels by hand. If you did not draw the images, please provide (a) a link to the source of the images or icons and their license / terms of use; or (b) written permission from the copyright holder to publish the images or icons under our CC BY 4.0 license. Alternatively, you may replace the images with open source alternatives. See these open source resources you may use to replace images / clip-art:

11) In the online submission form, you indicated that "All raw data will be made available on request." All PLOS journals now require all data underlying the findings described in their manuscript to be freely available to other researchers, either

1. In a public repository

2. Within the manuscript itself

3. Uploaded as supplementary information.

12) Please amend your detailed Financial Disclosure statement. This is published with the article. It must therefore be completed in full sentences and contain the exact wording you wish to be published.

3) If any authors received a salary from any of your funders, please state which authors and which funders.

13) Please ensure that the funders and grant numbers match between the Financial Disclosure field and the Funding Information tab in your submission form. Note that the funders must be provided in the same order in both places as well. Currently, "Simons Foundation" is missing from the Financial Disclosure field.

**Reviewers' comments:**

Reviewer's Responses to Questions

**Comments to the Authors:**

**Please note that one of the reviews is uploaded as an attachment.**

Reviewer #1: In the current manuscript, Kaushik et al. have been studying the function of PCDH15 in mechanosensory hair cells. Previous studies have shown that PCDH15 is expressed in three isoforms, which differ in their cytoplasmic domains and are generated by alternative splicing. It had also been shown that one of these isoforms, PCDH15-CD2, is a component of kinociliary links that connect the longest stereocilia of hair cells to the kinocilium. Furthermore, in PCDH15-CD2 mutant mice, planar cell polarity (PCP) of hair cells is disrupted leading to misorientation of the stereocilia bundle in the apical hair cell surface. Finally, studies from the laboratory presenting the current study have provided evidence that the CD2 cytoplasmic domain of PCHD15 is phosphorylated on tyrosine residues. The authors thus wanted to address the mechanism by which PCDH15-CD2 and its tyrosine phosphorylation affect PCP.

To conduct their studies, the authors generated mutant mice lacking the PCDH15-CD2 isoform or carrying mutations in the tyrosine residues within CD2. They demonstrated that CD2 tyrosine phosphorylation is not required for PCDH15-CD2 function in PCP, but they confirm earlier findings that PCP is disrupted in PCDH15-CD2 mutant mice. They then use immunolocalization studies for PCDH15-CD2 and other genes linked to PCP signaling such as GPSM2 and Gai to study effects of the PCDH15-CD2 mutation. They also generate PCDH15-CD2 and GPMS2 compound mutant mice to define potential genetic interactions between PCDH15-CD2 and genes linked to PCP signaling. They conclude that PCDH15-CD2 affects PCP independent from its function in the formation of kinociliary links.

Unfortunately, while the authors have done a lot of work, I do not find that the experiments support the conclusion put forward. Substantial more work would be necessary to establish what might be an interesting idea. There are also several technical limitations to the study:

1.The authors argue that PCDH15-CD2 is expressed at the base of kinocilia prior to the formation of kinociliary links. The immunohistochmical data are not convincing (Fig. 3). They use PCDH15-CD2-HA mice, which express an epitope tagged PCDH15-CD2 to reach the conclusions. They do not show non-HA controls. The PCDH15-CD2-HA mice carry several mutations in CD2, which could affect expression or localization. Also, more convincing pictures of HA staining would be required such as co-staining with markers for the base of the kinocilium and sagittal views of sectioned hair cell. There are also PCDH15 antibodies available from several laboratories to confirm the findings. Publications from the Barr-Gillespie laboratory show what kind of resolution can now be obtained to visualize subcellular distribution of proteins in hair cells. Finally, the staining data should be quantified.

2.The authors stain hair cells around E16.5-E18.5 to support the conclusion that labeling at the base of kinocilia is observed prior to the presence of kinociliar links. However, kinociliary links are already present at this age (even in the publication from the Richardson lab they cite, which is not the most convincing example, kinociliar links are present at this age, especially at the base and top of stereocilia). If the authors want to make this point, they would have to show that unlike what is published, they cannot find kinociliary links at an age when they observe PCP defects.

3.The authors show that polar Gai localization is affected in PCDH15-CD2 mice (Fig. 5A-C). This is interesting but may not be unexpected since PCP is affected in the mutants. Thus, the localization of PCP components might be perturbed. The problem is that we do not know what is cause and what is consequence of what. No experiment directly addresses a mechanistic link between PCDH15-CD2 and Gai.

4.The authors suggest that pericentrin localization may be affected in PCDH15-CD2 mutant mice (Fig. 4G,H). The images are not convincing and would need quantification. How do we know what is the mother/daughter centriole.

5.Intercrosses between PCDH15-CD2 and GPMS2 mutant mice suggest that the two genes genetically interact. The effect is particularly striking for the vestibule, where PCDH15-CD2/GPMS2 mutant mice have a far more pronounced hyperactivity defect than single mutants. However, vestibular hair bundles are not investigated. It should also be noted that there is a profound difference in the kinocilia of cochlear and vestibular hair cells. In the cochlea kinocilia in hair cells are transiently present during development, but in the vestibule they are maintained into adulthood where they are thought to be important for stereocilia stimulation by mechanical force. Thus, the vestibular impairment could indicate a functional defect in hair cells that have to do with coupling of the hair bundle to the overlying matrix to enable efficient force sensing. Or the stereocilia might have degenerated. Or any of several other possibilities. This would likely need a separate detail study to define mechanisms.

Reviewer #2: In this manuscript by Kaushik et al the contribution of protocadherin 15 (Pcdh15) towards stereociliary bundle development is evaluated using a unique conditional allele that produces a splicing-isoform specific knockout that can be converted to an epitope tagged and phosphorylation-deficient mutant. Using this line the authors expand upon prior characterizations of Pcdh15 and propose a model in which Pcd15-CD2 contributes to hair cell planar polarity. The KO has a striking impact on stereociliary bundle morphology in which the kinocilium is frequently separated from the stereocilia (kinicilium-stereocilia decoupling). As a result, the position of the kinocilium on the apical cell surface lacks the distinctive polarized localization to adjacent to the tallest stereocilia. In contrast, other aspects of hair cell intrinsic planar polarity and planar cell polarity are not disrupted in this mutant. For example, the localization of the intrinsic polarity factor Galpha i remains polarized with the protein enriched on the lateral edge of the apical hair cell surface. In addition, the rows of stereociliary bundles appear appropriately polarized and organized in rows of increasing height that are correctly positioned adjacent to Galpha i. Lastly, the polarized distribution of Core PCP proteins like Vangl2 is not disrupted in the Pdc15-Cd2 mutants. Altogether these results raise the question of how cell intrinsic planar polarity should be defined, and what phenotype criteria are needed to categorize it as disrupted. The intellectual merit of the manuscript would be improved if the authors discussed these questions and provided a clear argument as to why Pcd15-CD2 is contributing to intrinsic planar polarity as per the title rather than maintaining stereociliary bundle integrity by linking the kinocilium to the stereocilia.

Additional concerns

1)What is Pcdh15-CD2 and how is it different from Pdch15-CD1 and Pcdh15-CD-3? The introduction would benefit from the addition of an introduction to Pcdh15 alternative splicing and isoform function.

2)Figure 1: Since the kinocilium is decoupled from the stereocilia in the n38 mutants and the kinocilium and stereocilia seem to be differentially effected, the authors should make independent polarity measures for stereocilia polarization.

3)Line 244: Since the Preyer’s reflex is not applied to n38YF mice so it is not clear whether or not they can hear based on this assay. Without that result it is not accurate for this section of the manuscript to be titled “Tyrosine phosphorylation of the CD2 domain is not essential for Pcdh15 Function in mice” since the contribution phosphorylation towards hearing has not been tested.

4)Preyer’s reflex is not a robust or rigorous assay for auditory function and the authors should consider ABRs

5)Fig.7A: unlike Galpha I, the area of cell surface containing GPSM2 does not change in Pcdh15 KOs despite the colocalization of these proteins reported by others. This result is unexpected and should be discussed.

Minor concerns

1)It would be useful to report the frequency of circular bundles since they are not common in other figure panels and so that the may be compared to the frequency of other mutants like IFT88

2)Line 44: Hearing perception does not occur in the cochlea but rather in auditory cortex. This should be corrected.

3)Line 81: Since a hair cell only has one kinocilium this phenotype should be ‘kinocilium-stereocilia decoupling’

4)It would help your readers if the text stated that Pgk1-Cre was used to make a germline mutant particularly since Cre is more often used for cell or tissue-specific gene deletion

5)Figure 1 schematic: it is not obvious what parts of the protein are encoded by exon 38 or how alternative splicing could exchanges CD1 or CD3 for CD2

6)Fig8: The genotype nomenclature “ -/- “ that is used for the wild type Atoh1 allele is confusing since “ -/- “ typically indicates a null allele or knockout. Why not say Cre-negative or nothing at all? Similarly, the abbreviations ‘-ve’ and ‘+ve’ for positive and negative can be confusing. Why not just use HA-positive and HA-negative?

Reviewer #3: In this manuscript, Kaushik et al. aimed to understand the role of Pcdh15 in planar cell polarity (PCP) establishment of hair cells in the developing cochlea. They reported a Pcdh15-n38 knock-in mouse which was designed to tag the endogenous Pcdh15-CD2 isoform, but found that the inserted FLAG tag was undetectable, and the mice were phenotypically similar to Pcdh15-CD2 knockouts in terms of PCP and centriolar defects. Upon Cre-mediated recombination, their allele did express an HA-tagged, phospho-dead CD2 domain, which partly rescued PCP defects. These results and the immunodetection of CD2 using the HA tag suggested an early role of Pcdh15 in PCP establishment independent of its interaction with the kinocilium. Furthermore, the authors generated compound mutants with Pcdh15-n38 and Gpsm2, showing genetically synergistic effects in PCP. These results are original and important to further understanding the mechanism of PCP establishment in hair cells. The paper is overall technically sound, but some conclusions are not fully supported by the data, and some interpretations do not come through (detailed below). The writing can be improved to be more understandable for general audience, as some technical terms specific to the hair cell biology were not defined when they were first used. Some essential information such as the sample size, definition of axes in the figure, and abbreviations in the figure are missing.

1. Missing definition: CD2, bare zone, ve (Fig. 8), arrows in the figures.

2. Western blot in Fig. 1C should have a loading control. Also, the authors should give some explanations/speculations about why the n38 allele did not express before Cre recombination. Was is truly equivalent to CD2 knockout at the molecular level? Is it possible to characterize the transcript at the RNA level?

3. In Fig. 4GH, it is unclear what "right angles" mean. Quantification of the angles would make more sense.

4. Line 337, why is the interaction between Pcdh15 and Gpsm2 likely to be downstream? A synergistic effect could happen with a common upstream regulator. Line 414 said their localizations were independent, which is contradictory to Fig. 5, where the n38 mutation changed Gai localization.

5. In Fig. 8D, it seems that the HA-negative cells also had a significant rescue of n38 in kinocilia deviation, which needs explanation.

**Have all data underlying the figures and results presented in the manuscript been provided?**

Reviewer #1: Yes

Reviewer #2: Yes

Reviewer #3: **No: ** I did not find the raw data for plotting the graphs.

PLOS authors have the option to publish the peer review history of their article (what does this mean? ). If published, this will include your full peer review and any attached files.

**Do you want your identity to be public for this peer review?** For information about this choice, including consent withdrawal, please see our Privacy Policy .

Reviewer #1: No

Reviewer #2: No

Reviewer #3: No

**Figure resubmission:**

**Reproducibility:**

To enhance the reproducibility of your results, we recommend that authors of applicable studies deposit laboratory protocols in protocols.io, where a protocol can be assigned its own identifier (DOI) such that it can be cited independently in the future. Additionally, PLOS ONE offers an option to publish peer-reviewed clinical study protocols. Read more information on sharing protocols at https://plos.org/protocols?utm_medium=editorial-email&utm_source=authorletters&utm_campaign=protocol

---

## [Decision Letter · Decision Letter 1]

1 Jul 2025

PGENETICS-D-24-01537R1

Role of Pchd15 in the Development of Intrinsic Polarity of Inner Ear Hair Cells

PLOS Genetics

Dear Dr. Ladher,

Thank you for submitting your manuscript to PLOS Genetics. After careful consideration, we feel that it has merit but does not fully meet PLOS Genetics's publication criteria as it currently stands. Therefore, we invite you to submit a revised version of the manuscript that addresses the points raised during the review process.

Please submit your revised manuscript within 30 days Jul 31 2025 11:59PM. If you will need more time than this to complete your revisions, please reply to this message or contact the journal office at plosgenetics@plos.org. Please include the following items when submitting your revised manuscript:

We look forward to receiving your revised manuscript.

Kind regards,

Nandan Nerurkar

Academic Editor

PLOS Genetics

Fengwei Yu

Section Editor

PLOS Genetics

Aimée Dudley

Editor-in-Chief

PLOS Genetics

Anne Goriely

Editor-in-Chief

PLOS Genetics

**Journal Requirements:**

1) In the online submission form, you indicated that "All raw data will be made available on request". All PLOS journals now require all data underlying the findings described in their manuscript to be freely available to other researchers, either

1. In a public repository

2. Within the manuscript itself

3. Uploaded as supplementary information.

**Reviewers' comments:**

Reviewer's Responses to Questions

**Comments to the Authors:**

Reviewer #1: The authors have substantially revised the manuscript. While the data are interesting, I do not find that the data support the conclusion that “Pcdh15-CD2 couples kinocilia and stereocilia development through a mechanism independent of kinocilial links” (abstract). They also conclude that Pcdh15-CD2 is linked to Gai/Gpsm2 signaling, which in my opinion is not clearly shown.

Confirmative findings:

Using new mouse models, the authors confirm earlier findings that the Pcdh15-CD2 isoform is expressed in the kinocilium of hair cells and required for coupling of the kinocilium to stereocilia. They also confirm that hair cells of Pcdh15-CD2 mutant mice have a defect in the intrinsic polarity as revealed in the morphology of stereocilia bundles.

New findings:

1.Phosphorylation of Pcdh15-CD2 is not crucial for the development of a polar hair bundle

2.Localization of Gai is perturbed in Pcdh15-CD2 mutant mice consistent with a defect in intrinsic polarity

3. Pcdh15-CD2 mutations seem to genetically interact with Gpsm2 mutations in both the cochlea and vestibule (but see comment below).

Major critique points:

1.I do not see convincing evidence that the polarity defects cannot be explained by defects in the coupling of the stereocilia to the kinocilium via kinociliar links. The authors argue that Pcdh15-CD2 is expressed prior to the formation of kinociliary links and that polarity defects materialize prior to the presence of kinociliary links. To me the supporting data are not of sufficient quality and resolution to reach that conclusion.

2.The fact that Gai is mis-localized in Pcdh15-CD2 mutants could be a consequence of polarity defects without a direct mechanistic link.

3.The genetic interaction data are interesting but also do not provide direct evidence of a mechanistic link. Weakening two molecular pathways that run in parallel (e.g.: kinocilia-stereocilia cohesion via Pcdh15-CD2, and molecular pathways regulating the polarity of the cytoskeleton) could explain stronger phenotypes. In addition, there is an issue with the quantification of the genetic interaction data. Fig. 6E lacks the proper control, comparison of double mutants to Pcdh15-n38/n38 mutants. Pcdh15-n38/n38 mutants are shown in Fig. 1H and it seems that kinocilia mislocalization is similarly affected in Pcdh15-n38/n38 mutant mice (Fig. 1H) compared to Pcdh15-n38/n38;Gspm2-/- mice (Fig. 6E). Perhaps the effect is slightly enhanced but since there is no direct statistical comparison, it is hard to be sure. Maybe a stronger phenotype shows up for fragmentation of hair bundles in double mutants (Fig. 6F), but this has not been quantified for Pcdh15-n38/n38 mutants. Likewise, the proper control (Pcdh15-n38/n38 mutant mice) is lacking for the analysis of the vestibular phenotype. Of note, it has previously been shown that Pcdh15-CD2 mutant mice have a vestibular defect.

Additional points:

1.Introduction: “hearing is sensed “ should be correcte. We do not sense hearing, we sense sound.

2.FM-143 uptake to study mechanotransduction: controls a lacking to show that uptake is via mechanotransduction channels and not by endocytosis. A control experiments should be performed: FM-143 uptake in the presence of channel blockers.

3.The authors conclude that the hearing defects (Preyers reflex) cannot be explained by a defect in mechanotransduction. They cannot conclude this. The FM-143 experiments were carried early postnatally, the hearing tests at P60. By that age, mechanotransduction could be affected.

4.The authors should measure ABRs, maybe at P21 and P60. Preyers reflex is insufficient to reveal if there are significant defects in auditory thresholds (maybe the phosphorylation mutants have a hearing defect not revealed by the crude assay).

5.In Fig.3A, dots for HA staining are all over the tissue, sometimes with multiple dots per cells. Staining seems to be present in hair cells and supporting cells. Is this correct? Also, why not co-stain in Fig. 3A with kinociliary markers to solidify the conclusion that the staining is in kinocilia. And show controls, in parallel with mice not expression an HA-tagged protein to exclude background. I appreciate that staining controls are shown in the response letter, but they should be part of the main figure.

Reviewer #2: While the authors have made selective improvements to the manuscript, they have left my two primary concerns unaddressed:

1. What constitutes intrinsic polarity, and what phenotype criteria must be met to categorize it as disrupted?

2. Why should Pcd15-CD2 be interpreted as a contributor to intrinsic planar polarity—as the title implies—rather than as a mediator of stereociliary bundle cohesion via kinocilium–stereocilia linkage?

I raised these points because I remain unconvinced that Pcd15-CD2 plays a direct role in establishing intrinsic hair cell planar polarity. That said, the authors should be given room to explain their rationale and clarify why they believe Pcd15-CD2 is not regulating kinocilium–stereocilium adhesion, with only indirect effects on cell orientation. They have not done so.

To their credit, they now include additional measures of hair cell orientation based on stereociliary alignment and compare them with kinocilium-based measures. They demonstrate that these two methods reveal different phenotype severities—an important observation that underscores how divergent interpretations can result from the selection of different phenotyping criteria. This discrepancy warrants explicit discussion, particularly regarding how "disruption" is defined and by what criteria. It also reinforces my own interpretation that the primary function of Pcd15-CD2 is kinocilium:stereocilia linkage and that any impacts on intrinsic hair cell polarity are indirect.

Regarding my second point, it's notable—if not puzzling—that the adhesive role of Pcd15-CD2 receives little to no attention in the discussion. Given the protein’s known function in kinocilium–stereocilium cohesion, the authors should clearly articulate why they are interpreting it to have a direct role in intrinsic polarity regulation instead, and why readers like myself should reconsider their existing expectations. A few well-placed sentences in the Discussion section would assuage my contrary interpretation.

Reviewer #3: The authors did a good job addressing my questions.

One minor point: "polarity" in the Keywords section was misspelt.

**Have all data underlying the figures and results presented in the manuscript been provided?**

Reviewer #1: Yes

Reviewer #2: Yes

Reviewer #3: Yes

PLOS authors have the option to publish the peer review history of their article (what does this mean? ). If published, this will include your full peer review and any attached files.

**Do you want your identity to be public for this peer review?** For information about this choice, including consent withdrawal, please see our Privacy Policy .

Reviewer #1: No

Reviewer #2: No

Reviewer #3: No

**Figure resubmission:**
---

## [Editor Report · Decision Letter 2]

1 Aug 2025

Dear Dr Ladher,

We are pleased to inform you that your manuscript entitled "Role of Pcdh15 in the Development of Intrinsic Polarity of Inner Ear Hair Cells" has been editorially accepted for publication in PLOS Genetics. Congratulations!

Yours sincerely,

Nandan Nerurkar

Academic Editor

PLOS Genetics

Fengwei Yu

Section Editor

PLOS Genetics

Aimée Dudley

Editor-in-Chief

PLOS Genetics

Anne Goriely

Editor-in-Chief

PLOS Genetics

Comments from the reviewers (if applicable):

**Data Deposition**

http://datadryad.org/submit?journalID=pgenetics&manu=PGENETICS-D-24-01537R2

**Press Queries**

---

## [Editor Report · Acceptance letter]

PGENETICS-D-24-01537R2

Role of Pcdh15 in the Development of Intrinsic Polarity of Inner Ear Hair Cells

Dear Dr Ladher,

We are pleased to inform you that your manuscript entitled "Role of Pcdh15 in the Development of Intrinsic Polarity of Inner Ear Hair Cells" has been formally accepted for publication in PLOS Genetics! Your manuscript is now with our production department and you will be notified of the publication date in due course.

With kind regards,

Zsofia Freund

PLOS Genetics

On behalf of:
